# A comparative analysis of planarian genomes reveals regulatory conservation in the face of rapid structural divergence

Mario Ivanković [1,7], Jeremias N. Brand [1,7], Luca Pandolfini [2], Thomas Brown [3], Martin Pippel[3], Andrei Rozanski [1], Til Schubert[1], Markus A. Grohme [3], Sylke Winkler [3], Laura Robledillo[4], Meng Zhang [4], Azzurra Codino [2], Stefano Gustincich[2], Miquel Vila-Farré [1], Shu Zhang[5], Argyris Papantonis [5], André Marques [4] & Jochen C. Rink [1,6] ✉

The planarian *Schmidtea mediterranea* is being studied as a model species for regeneration, but the assembly of planarian genomes remains challenging. Here, we report a high-quality haplotype-phased, chromosome-scale genome assembly of the sexual S2 strain of *S. mediterranea* and high-quality chromosome-scale assemblies of its three close relatives, *S. polychroa*, *S. nova*, and *S. lugubris*. Using hybrid gene annotations and optimized ATAC-seq and ChIP-seq protocols for regulatory element annotation, we provide valuable genome resources for the planarian research community and a first comparative perspective on planarian genome evolution. Our analyses reveal substantial divergence in protein-coding sequences and regulatory regions but considerable conservation within promoter and enhancer annotations. We also find frequent retrotransposon-associated chromosomal inversions and interchromosomal translocations within the genus *Schmidtea* and, remarkably, independent and nearly complete losses of ancestral metazoan synteny in *Schmidtea* and two other flatworm groups. Overall, our results suggest that platyhelminth genomes can evolve without syntenic constraints.

Evolution acts on genomic changes to bring about the diversity of life. For example, single nucleotide changes in coding gene sequences duplicate the goldfish tail fin[1] or cause nose loss in humans[2]; changes in gene regulatory regions are associated with profound evolutionary body plan changes[3,4], e.g., limb loss in snakes[5], and gene loss is emerging as an important mechanism in trait evolution[6,7]. On the other hand, the rapidly increasing number of sequenced genomes indicates that genome structure may also be evolutionarily constrained.

Synteny, the association of genes on a chromosome or linkage group, is deeply conserved among animals, with the Metazoan Ancestral Linkage Groups (MALG) being conserved in numerous animal phyla, including Sponges, Cnidarians, and Bilateria[8–10]. Other groups, including Nematodes and Drosophilids, have lost this ancestral synteny, but have replaced it with group-specific linkage groups[11,12]. The finding that the arrangement of genes in the genome at the megabase scale is important for gene regulation[13] provides a rationale for the evolutionary conservation of synteny. Indeed, molecular studies have shown that gene regulation is influenced by hierarchical levels of chromatin organization[14] and that the modulation of chromatin organization can lead to genetic disease[15,16], cancer[17,18], or even the origin of

[1]Department of Tissue Dynamics and Regeneration, Max Planck Institute for Multidisciplinary Sciences, Göttingen, Germany. [2]Center for Human Technologies, Non-coding RNA and RNA-based therapeutics, Istituto Italiano di Tecnologia, Genova, Italy. [3]Max Planck Institute of Molecular Cell Biology and Genetics, Dresden, Germany. [4]Department of Chromosome Biology, Max Planck Institute for Plant Breeding Research, Cologne, Germany. [5]Institute of Pathology, University Medical Center Göttingen, Göttingen, Germany. [6]Faculty of Biology und Psychology, Georg-August-University Göttingen, Göttingen, Germany. [7]These authors contributed equally: Mario Ivanković, Jeremias N. Brand. ✉e-mail: jochen.rink@mpinat.mpg.de

evolutionary novelty[19–21]. However, not all taxa exhibit consistent features of genomic organization or their significance remains unclear[22]. This raises the possibility that certain taxonomic groups may display specific patterns of genome evolution and that the analysis of under-sampled clades may reveal novel patterns.

Planarians are an example of a large and poorly sequenced group of animals. As an order (Tricladida) within the diverse and species-rich phylum Platyhelminthes (flatworms), planarians are being studied for their capacity for whole-body regeneration and their abundant adult pluripotent stem cells[23,24]. Although the planarian model species *Schmidtea mediterranea* was among the first cohort of Sanger-sequenced genomes, the resulting assembly was highly fragmented[25]. Significant assembly contiguity was only achieved with the advent of long-read sequencing[26], which has been extended recently to chromosome-scale with Hi-C scaffolding[27]. The strong compositional bias (> 70% A/T), abundant repeats including giant > 30 kb Burro retroelements, and inbreeding-resistant heterozygosity[26,28] associated with a large chromosomal inversion on Chromosome 1[27] are some of the reasons why the *S. mediterranea* genome remains an assembly challenge. The extent to which these peculiarities are species-specific or general features of planarian genomes remains unknown, as the other available planarian genome assemblies are highly fragmented[29–31]. Parallel functional genomics efforts using ATAC-seq and ChIP-seq have initiated the annotation and analysis of genomic features of *S. mediterranea* and have provided first insights into the function of epigenetic regulators and gene regulatory elements[32–40]. Furthermore, the annotation of gene regulatory elements across the genome would benefit from the assessment of evolutionary sequence conservation in related species. Additional planarian genome assemblies are therefore essential to gain such a comparative perspective and to start exploring the genetic basis of the rich phenotypic biodiversity within the group[41].

Here, we present four high-quality genome assemblies of *S. mediterranea* and its three close relatives, *S. polychroa*, *S. nova*, and *S. lugubris*, including haplotype-phasing in the case of the model species. The chromosome-scale assemblies and substantially improved gene annotations provide important model system resources for the planarian research community. Using improved ATAC-seq and ChIP-seq protocols, we further identify and annotate regulatory elements conserved across the genus and represent attractive targets for functional investigations. In contrast, we find that the genome structure is poorly conserved within the genus and, interestingly, that *Schmidtea* and the free-living early-branching flatworm *Macrostomum hystrix* have independently lost the Metazoan Ancestral Linkage Groups. Altogether, our study provides a first comparative perspective on the *S. mediterranea* genome and indicates that synteny may not constrain the structural evolution of planarian genomes and those of other flatworms.

## Results

### *Schmidtea mediterranea* reference genome improvements
The current *S. mediterranea* reference genome (dd_Smes_g4[26]) is a haploid consensus assembly containing 481 contigs and its recent scaffolding (referred to here as schMedS2) has revealed substantial haplotypic differences, especially on Chromosome 1[27]. To generate a haplotype-phased assembly and to close the remaining sequencing gaps, we re-sequenced the *S. mediterranea* genome using Pacific Biosciences' HiFi reads and used Hi-C for scaffolding. The new assembly, designated S3, consists of two pseudo-haplotypes: S3h1 and S3h2, and a merged version of the two, referred to as S3BH for "S3 both haplotypes". The S3h1 (662 contigs) and S3h2 (432 contigs) assemblies are slightly larger than the previous dd_Smes_g4 assembly (Table 1, Supporting Information: Section 1.1). The N50 values of 270 Mb and 269 Mb indicate high contiguity with 95% and 96% of the total assembly contained in the four largest scaffolds that match the known

karyology in size and number (1n = 4, Table 1). The Hi-C contact maps indicate high contiguity in both phases, thus justifying the designation "chromosome-scale" assembly (Fig. 1a). Nevertheless, not all scaffolds are capped by telomere repeats, and 662 and 432 unincorporated contigs remain for S3h1 and S3h2, respectively). The largest fraction of unincorporated contigs comprises various repeat sequences (satellite DNA, telomere repeats, rRNA clusters). Still, some contigs contain annotated genes (S3h1: 505 genes on 216 contigs, S3h2: 509 genes on 197 contigs), pointing towards remaining localized assembly ambiguities.

We evaluated the assemblies base-pair accuracy and completeness using a Merqury analysis of four independent short-read gDNA datasets (see Methods). We found that each haplotype was as complete as the previous assemblies, but both were at least an order of magnitude more accurate (Fig. 1b). Naturally, S3BH showed similar high accuracy but was also more complete, representing ~98% of the short-read datasets (Fig. 1b, Supporting Information: Section 1.1). Genome completeness assessment using Benchmarking Universal Single-Copy Orthologs (BUSCO) again showed that S3h1 and S3h2 were more complete than the previous assemblies[27] (Supporting Information: Section 1.1). Since S3h1 scored slightly higher in both metrics, we chose S3h1 as our focal assembly for all the following analyses.

To independently assess the long-range contiguity of our assembly, we aligned our phased assemblies with schMedS2 and found a high degree of structural agreement between the two independent scaffolding efforts (Supporting Information: Section 1.2). Additionally, the S3 assembly successfully captured prominent repeat regions on all chromosomes that were absent in previous assemblies, which likely contributed to the slightly larger assembly size. Repetitive regions were often > 1 Mb (e.g., Chromosome 4, Fig. 1c) and were comprised of nested tandem repeats (Fig. 1d), which could be reconstructed due to the high base-pair accuracy of HiFi reads (Fig. 1b). Besides the three inversions on Chromosome 1 (Fig. 1e) and one inversion on Chromosome 2 that were described previously[27], we detected an inversion on Chromosome 4. Heterozygosity was largely restricted to the inversion on Chromosome 1 and to the central region of Chromosome 2 containing the inversion (Fig. 1e), again in agreement with prior findings[27]. The mapping of the Hi-C data onto the diploid assembly (S3BH) revealed an abundance of unique mapping reads to regions with high heterozygosity within the inversions of Chromosome 1, 2, and 4. Therefore, the phasing likely accurately represents the haplotype divergence in these regions (Supporting Information: Section 1.3). In contrast, the much lower haplotype difference in non-inverted parts of Chromosome 2 and most of Chromosome 3 is likely due to the extensive inbreeding of our genome strain (> 18 generations) and more frequent recombination in non-inverted regions[26,27]. The gene distribution across the chromosomes was largely uniform, lacking the typical reduction in gene density toward the centromere (Fig. 1e). A notable exception is Chromosome 4, which was marked by an increase in transposon density and a concurrent decrease in gene density at the metacentric chromosome (Fig. 1e). Overall, the S3 phased genome assembly represents significant improvements over previous assemblies in terms of accuracy, completeness, and assembly contiguity and thus a strategic community resource for the analysis of gene function in the model species *S. mediterranea*.

### *Schmidtea mediterranea* genome annotation improvements
We further sought to complement the new genome assembly with high-quality gene annotations. Encountering increasingly diminishing returns on investment with our previous de novo gene prediction approaches[42], we developed a new hybrid approach that merges Oxford Nanopore long-reads (ONT), Illumina short-reads, and 3P-seq of transcription termination sites (TTS) data[43] with genome-guided transcript assembly, thus leveraging the benefits of direct gene

**Table 1 | Summary statistics for the genome assemblies and annotations of the four *Schmidtea* species**

| Species | *S. mediterranea* | | *S. polychroa* | *S. nova* | *S. lugubris* |
|---|---|---|---|---|---|
| Assembly | schMedS3h1 | schMedS3h2 | schPol2 | schNov1 | schLug1 |
| **Genome assembly statistics** | | | | | |
| # contigs | 662 | 432 | 1013 | 283 | 320 |
| Total length (bp) | 840,173,815 | 819,865,861 | 781,290,622 | 1,251,382,582 | 1,499,048,548 |
| GC (%) | 29.59 | 29.59 | 28.11 | 28.13 | 27.87 |
| N50 (bp) | 270,168,396 | 268,961,546 | 189,691,935 | 455,729,997 | 498,167,912 |
| chromosome scaf. (%) | 95 | 96 | 89 | 98 | 99 |
| unplaced (Mb) | 42 | 32.8 | 85.9 | 25 | 15 |
| **Gene annotation statistics** | | | | | |
| Number of genes | 58,735 | 58,551 | 41,915 | 41,539 | 48,029 |
| Median gene length (bp) | 1113 | 1104 | 1137 | 1005 | 1039 |
| Shortest gene (bp) | 83 | 83 | 68 | 85 | 81 |
| Longest gene (bp) | 379,876 | 379,876 | 399,802 | 422,721 | 513,405 |
| Total gene length (bp) | 395,812,229 | 391,333,147 | 360,601,815 | 508,048,891 | 610,689,353 |
| Number of transcripts | 86,102 | 85,741 | 60,887 | 57,075 | 66,510 |
| Median transcript length (bp) | 936 | 930 | 950 | 842 | 860 |
| Shortest transcript (bp) | 83 | 83 | 68 | 85 | 61 |
| Longest transcript (bp) | 76,733 | 94,201 | 32,100 | 36,224 | 28,220 |
| Total transcript length (bp) | 111,559,651 | 110,732,822 | 79,151,531 | 68,678,530 | 79,879,797 |

Given is the number of contigs in the assembly, assembly length, GC content, N50 values, the percentage of assembly that is on chromosome scaffolds, and the number of bases that are not placed on a chromosome. For the annotation, the number of genes and transcripts and their length and span are indicated.

isoform evidence with the base pair accuracy of our genome assembly (Fig. 1f). We generated separate annotation sets for both haplotypes and further designated high-confidence (*hconf*) transcripts based on Open Reading Frame (ORF) length and/or a minimum coverage threshold (see Methods). With 58,739 and 58,551 gene loci and 21,401 and 21,310 *hconf* gene loci in S3h1 and S3h2, respectively, these annotations are in range with previous *S. mediterranea* gene number estimates[26,42,44]. To analyze the overall quality of the new annotations, we carried out systematic benchmarking comparisons against previous transcriptomes or *S. mediterranea* gene model predictions. We first assessed BUSCO representation and completeness. S3BH had the highest number of complete BUSCOs (789) and the fewest missing BUSCOs (134) of all the annotations tested. Interestingly, the transcriptome or gene model-based BUSCO scores were consistently better than genome-mode BUSCO assessments (Fig. 1g, Supporting Information: Section 1.1), indicating that the detection method used by BUSCO is sub-optimal for planarian genomes. Moreover, the comparatively high number of "missing" BUSCOs reflects a high proportion of genuine gene losses in planarians (see below) and highlights the need for a group-specific BUSCO set. As a further completeness measure, we analyzed the representation of 1075 *S. mediterranea* genes available from NCBI GenBank. S3BH again was more complete, with 1056 genes represented compared to 1004 in the current community reference, the dd_v6 transcriptome (Fig. 1h). In addition, 3 of the 19 transcripts "missing" in the S3BH annotations were false positives (e.g., mitochondrial transcripts, Supporting Information: Section 1.4), thus yielding overall annotation completeness > 98%. Similarly, when quantifying the mappability of published RNA-seq datasets as a global measure of gene annotation completeness, S3BH also outperformed the existing *S. mediterranea* gene annotations (Fig. 1i) inclusive of recently expanded gene annotation sets[38,45].

To assess the specificity of our gene annotations, we manually inspected and compared the representation of 96 genes amongst the different annotations. The test set consisted of often lowly expressed signaling pathway components and 50% random genes to provide an unbiased representation of planarian genes. Each gene in the dataset was scored for the presence or absence in the respective annotation and for commonly encountered gene model errors, including truncated, fragmented, frame-shifted, or chimeric transcript predictions (Fig. 1j). Overall, the S3 annotations performed best, containing models of all test genes and the highest proportion of error-free gene models with intact ORF representations of 93/96 test genes. The identical scores of the "all" versus "high confidence" S3 annotations reflect the inclusion of all 96 test genes in the "high confidence" category (see above). Although the S3 predictions still harbor a low proportion of chimeric, truncated, or frame-shifted transcripts (see Discussion and Supporting Information: Section 1.5), they nevertheless represent an improvement over the current annotations with only 78/96 error-free transcript representations and a higher proportion of fragmented transcripts. The S3 annotations, therefore, represent the most complete and accurate gene annotations of the model species *S. mediterranea* to date.

## Promoter and enhancer annotations in the *Schmidtea mediterranea* genome

The annotation of cis-regulatory elements is a further critical element in understanding the biology of an organism. To explore and annotate regulatory elements in the S3 genome assembly, we first identified accessible chromatin regions using ATAC-seq. Due to the high nuclease activity and abundant polysaccharides (mucus components) in planarian tissue[26], we modified an existing Omni ATAC-seq protocol[46] to minimize clumping of nuclei and to deplete free and mitochondrial DNA contamination (Supporting Information: Section 2.1, see "Methods" section). To annotate regions of accessible chromatin, we applied our protocol to whole intact (wt) or x-irradiated (x-ray) individuals of the extensively studied asexual strain of *S. mediterranea* and generated a high-confidence ATAC-seq peak set consisting only of peaks present in at least three biological replicates (Fig. 2a). The distribution of nucleosome-free fragments and progressively fewer mono-, di-, and trinucleosomal fragments in our ATAC-seq libraries[47] (Fig. 2b), the typical transcription start site (TSS) enrichment profiles of nucleosome-free and mononucleosomal fragments (Fig. 2c) and the size distribution of the ATAC-seq peaks (Fig. 2d) together indicate the high quality of our ATAC-seq data, as well as the TSS annotations of the S3 gene models. The merged peak set comprises 55,585 peaks with a mean length of 668 bp (Fig. 2d, Supplementary Data 1).

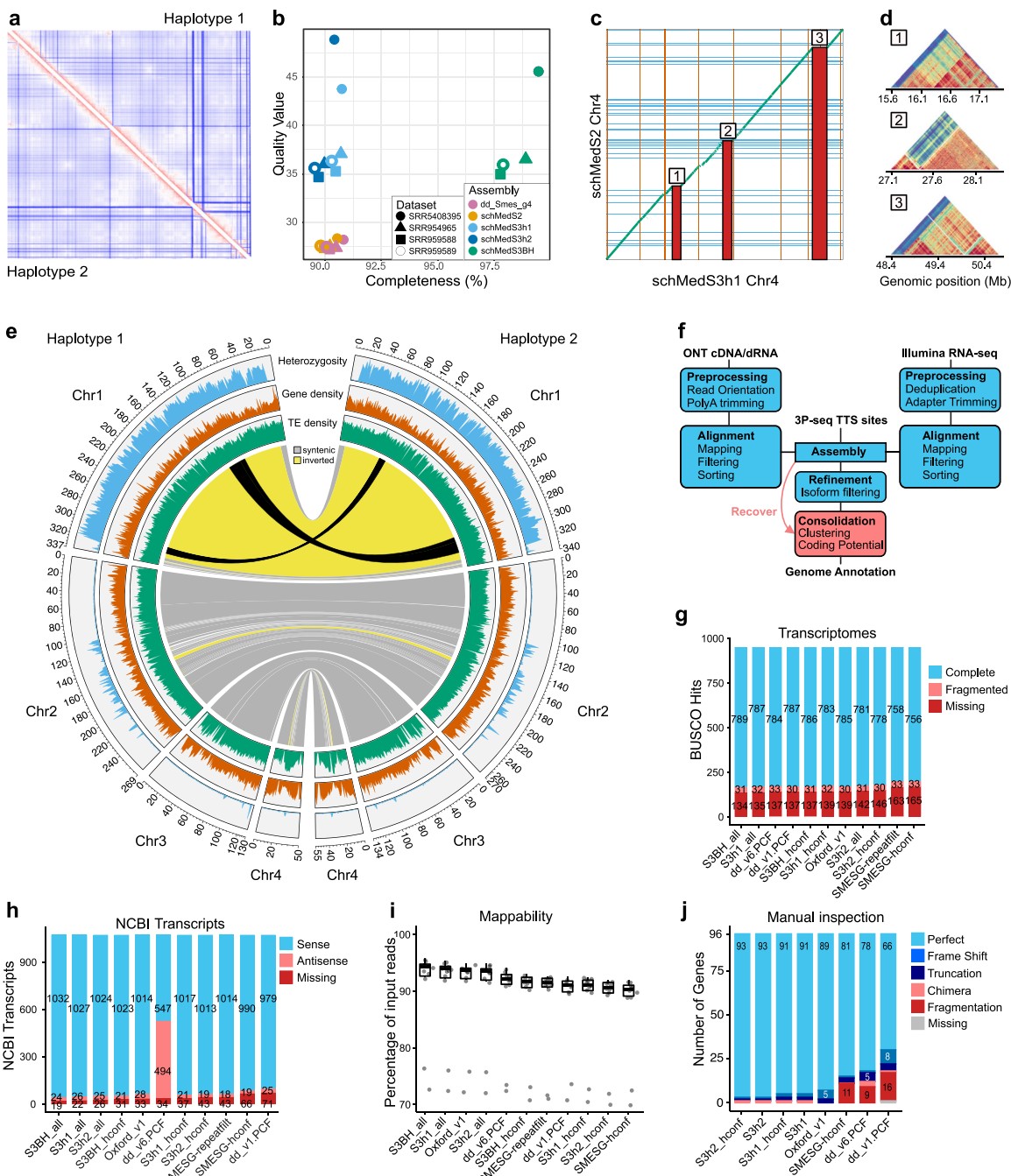

**Fig. 1 | Quality control metrics and description of the *S. mediterranea* genome and annotation. a** Hi-C contact map of the reads used for scaffolding on S3h1 (upper right) and S3h2 (lower left), showing high contact intensity in red and low contact intensity in blue. **b** Results of a Merqury analysis using four Illumina shotgun datasets not used for the assembly. **c** Dotplot representing a whole genome alignment between Chromosome 4, inferred with minimap2, of the previously scaffolded assembly (schMedS2) on the y-axis and the genome in this study (schMedS3h1) on the x-axis. Blue lines indicate scaffold gaps in schMedS2 and red lines indicate scaffold gaps in schMedS3h1. Numbered red bars indicate alignment gaps >1 Mb, which contain highly repetitive satellite DNA absent in the previous assembly. **d** Self-similarity heatmap, calculated with stained glass, of the numbered gaps in **c** showing their high self-similarity, typical of centromeric or pericentromeric repeats. **e** Comparison between the two pseudohaplotypes of schMedS3. The chord diagram in the center indicates synteny regions (grey) and inversions (yellow) between the haplotypes. The black ribbons within the large inversion in

Chromosome 1 indicate the contained smaller inversion. Density plots in the outer three circles show the distribution of transposable elements (TE), genes, and heterozygosity. **f** Representation of the hybrid gene annotation workflow. **g–i** Completeness comparison of benchmarked annotations using BUSCO (**g**), using the 1054 *S. mediterranea* transcripts deposited in GenBank (**h**), and using the mappability of 13 publicly available RNA-seq datasets (**i**). Box plots show the interquartile range (IQR), with whiskers extending to 1.5 times the IQR. **j** ORF integrity comparison of benchmarked annotations by manual inspection of 96 gene models for indicated error categories. The scores reflect only the best-predicted transcript/locus/benchmarked annotation. **g–j** Benchmarked gene annotations: S3h1, S3h2, S3BH: this study; dd_v1 the non-stranded dd_Smes_v1 assembly of the sexual strain of *S. mediterranea*[42]; dd_v6 the dd_Smed_v6 assembly of the asexual strain of *S. mediterranea*[42]; SMESG the gene prediction on basis of the previous dd_smes_g4 *S. mediterranea* genome assembly[42]; Oxford_v1 a composite annotation of[38,45], and SMESG. Source data are provided as a Source Data file.

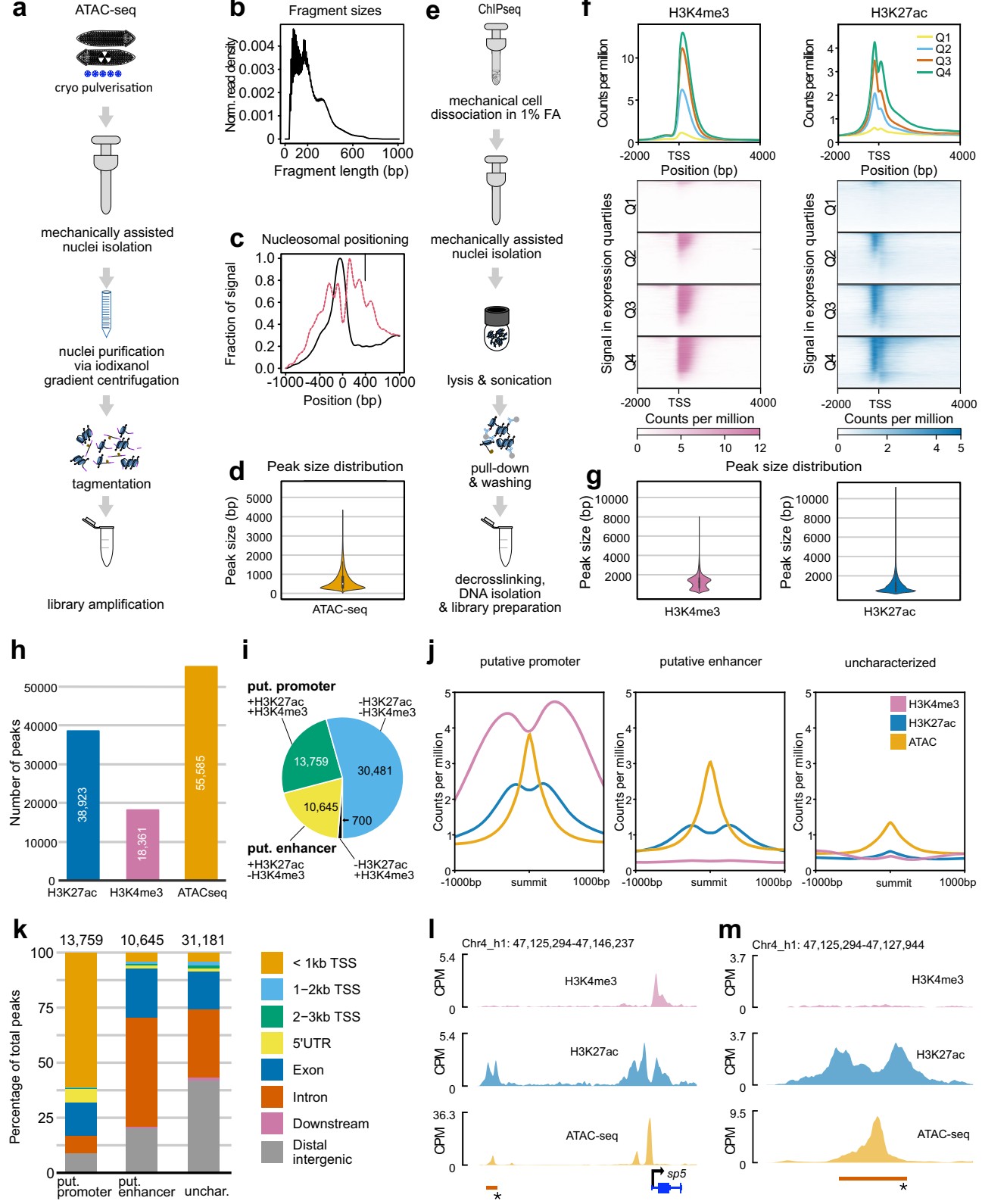

To further sub-categorize these accessible chromatin regions, we leveraged the known enrichment of specific histone marks at gene regulatory sequences (reviewed in ref. 48). To do so, we developed a ChIP-seq protocol that utilizes isolated nuclei from fixed tissue (Fig. 2e). As shown in Additional File 2: Section "*Schmidtea mediterranea* genome annotation", our protocol yielded adequate signal-to-noise ChIP-seq signals with both the

H3K4me3 (TSS and gene body of actively transcribed genes[49]) and H3K27ac (enhancers and TSS[50]) marks. In addition, the H3K27ac and H3K4me3 signals close to the TSS showed the expected distribution when stratified by gene expression levels (Fig. 2f). In total, our ChIP-seq dataset consists of 18,361 H3K4me3 peaks with a mean size of 1184 bp and 38,923 H3K27ac peaks with a mean size of 891 bp (Fig. 2g, h; Supplementary Data 2, 3).

**Fig. 2 | Profiling of the chromatin regulatory landscape in *Schmidtea mediterranea* using ATAC-seq and ChIP-seq. a** Schematic illustration of the ATAC-seq library generation workflow. **b** Representative fragment size distribution, with peaks at 100 and 200 bp reflecting nucleosome-free and mononucleosome-bound fragments. **c** Representative TSS enrichment analysis, revealing the expected enrichment of nucleosome-free region fragments (NFR, black) at the transcription start sites (TSS), flanked by enrichments of mononucleosomal fragments (mono, red). **d** Genome-wide peak size distribution of the 55,585 ATAC-seq peaks used in the subsequent analysis. **e** Schematic illustration of the ChIP-seq library generation workflow. **f** Genome-wide average TSS-centered coverage profiles and heatmaps of H3K27ac and H3K4me3 ChIP-seq read distributions, stratified by gene expression level quartiles. **g** Genome-wide peak size distribution of the 18,361 H3K4me3 and 38,923 H3K27ac ChIP-seq peaks. **h** Bar graph representation of the total number of H3K27ac, H3K4me3, and ATAC-seq peaks. **i** ATAC-seq peak categorization into putative promoters, putative enhancers, and uncharacterized peaks on basis of intersection with the two histone marks. **j** Averaged profiles of the H3K4me3 and H3K27ac ChIP-seq signal centered on the three predominant peak categories defined in **h**. **k** Stacked bar plots showing the distribution of the putative promoters, enhancers, and uncharacterized ATAC-seq peaks in relation to the high-confidence gene annotations. **l** Genome browser snapshot of the *sp5* gene locus, showing the exon/intron representation of *sp5* (blue, bottom right), a H3K4me3[+]/H3K27ac[+] promoter overlapping with the TSS and a H3K4me3[-]/H3K27ac[+] putative enhancer (red bar, asterisk) further upstream. **m** Close-up of the putative enhancer in l. Boxplots in **b** and **d** show the interquartile range (IQR), with whiskers extending to 1.5 times the IQR. Source data are provided as a Source Data file.

Intersecting our ChIP-seq peak sets with the 55,585 ATAC-seq peaks revealed a significant co-occurrence of ATAC-seq signal and both H3K4me3 signal (permutation test; 10,000 permutations, observed overlap: 15,465, permuted overlap: 2251, Z-score: 15,465, *p*-value < 0.0001, Fig. 2i) and H3K27ac signal (permutation test; 10,000 permutations, observed overlap: 25,550, permuted overlap: 4017, Z-score: 25,550, *p*-value < 0.0001, Fig. 2i). In line with the conserved functions of the examined histone marks, we designated the 13,759 ATAC-seq peaks with associated H3K27ac and H3K4me3 as 'putative promoters' and the 10,645 ATAC-seq peaks with only H3K27ac signal as 'putative enhancers'. The remaining 30,481 ATAC-seq peaks without H3K27ac or H3K4me3 signal and 700 ATAC-seq peaks with only H3K4me3 signal were designated 'uncharacterized accessible chromatin' (Fig. 2i; see Supplementary Data 1 for details on the classification of each called ATAC-seq peak). Consistent with this functional categorization, we found that 'putative promoters' collectively displayed a sharp ATAC-seq peak centered within "valleys" of both H3K4me3 and H3K27ac signals (Fig. 2j, Supporting Information: Section 2.3) and that > 61.4% were located within 1 kb upstream of a TSS annotation (Fig. 2k). In contrast, 'putative enhancers' collectively displayed the ATAC-seq peak in a "valley" of surrounding H3K27ac signal (Fig. 2i, Supporting Information: Section 2.3) and 80% were either intronic (49.2%), exonic (22.6%), or otherwise associated with a gene model (Fig. 2k). As expected, "Uncharacterized" ATAC-seq peaks lacked histone mark enrichment and displayed generally lower ATAC-seq signals (Fig. 2j). The *sp5* gene locus (Fig. 2l, m) illustrates the distribution of chromatin mark features, specifically a putative promoter immediately upstream of the TSS and a putative enhancer ~3 kb upstream of the TSS. In total, our whole-animal ATAC-seq/ChIP-seq approach annotated 13,759 putative promoter and 10,645 putative enhancer sequences. Given the 21,401 genes in the S3h1 *hconf* annotations (Supplementary Data 4), these are likely to be enriched for regulatory regions of constitutively expressed genes or those active in the most abundant cell types.

## Genomes and annotations for *S. polychroa*, *S. nova*, and *S. lugubris*

To orthogonally verify our peak annotations, we turned to the principle that the sequences of important regulatory elements are often conserved over evolutionary time[51]. Since multiple lines of evidence indicate unusually high sequence divergence within planarians and between flatworms in general[41,52–55], we sequenced the genomes of *S. mediterranea's* three closest relatives, *S. polychroa*, *S. nova*, and *S. lugubris* (Fig. 3a). All sequenced strains were diploid and displayed the expected karyotypes with 3 or 4 chromosomes[56–59]. The Hi-C maps of the assemblies indicated similar scaffolding as for *S. mediterranea* (Fig. 3b). In addition, the BUSCO scores (Fig. 3c) suggested a comparable completeness to the *S. mediterranea* S3 assembly (Fig. 1g). Interestingly, the assemblies of *S. nova* (1251 Mb) and *S. lugubris* (1499 Mb) were substantially larger than those of *S. mediterranea* (840 Mb) and *S. polychroa* (781 Mb) (Table 1).

To annotate the new genomes, we again used our hybrid transcriptome assembly strategy (Fig. 1f) yet without 3P-seq TTS evidence and coverage-based "high confidence" filter due to the lack of extensive RNA-seq data for these species. The annotation statistics indicated gene numbers similar to those of *S. mediterranea* (Table 1, Supplementary Data 4). Additionally, the BUSCO annotation completeness was improved compared to the run in genome mode and achieved comparable results to the *S. mediterranea* assembly (Fig. 3c, Supplementary Data 5, 6). Nevertheless, we identified 124 BUSCOs that were consistently missing across all four high-quality genomes, and 91 of these could also not be detected in four parasitic flatworms (see below), thus providing a further illustration of the previously noted substantial gene loss in planarians and other flatworms[26] (Supplementary Data 5, 6). The analysis of four-fold degenerate site divergence revealed considerable sequence divergence between the 4 *Schmidtea* species. *S. polychroa* differed from *S. mediterranea* by 0.3 substitutions per site, a distance analogous to that between humans and horses[60]. Both *S. nova* and *S. lugubris* show a divergence to *S. mediterranea* of ~0.6 substitutions at four-fold degenerate site, similar to the distance between humans and shrews[60] (Fig. 3a). Overall, our additional high-quality genome assemblies and annotations provide an interesting comparative perspective on the *S. mediterranea* genome, especially because the four assemblies are considerably more distant than one might expect of close sister species in other taxonomic branches.

## Conservation of gene regulatory regions

To explore the conservation of putative *S. mediterranea* gene regulatory elements in the other *Schmidtea* genomes, we collected ATAC-seq data in *S. polychroa*, *S. lugubris*, and *S. nova* under wt and x-ray conditions, allowing us to simultaneously assess genome sequence and chromatin accessibility conservation as proxies for functional conservation. The quality control analysis of all ATAC-seq data in the other *Schmidtea* species confirmed the robustness and species-independence of our protocol (Fig. 3d, e). Merging of the replicates and biological conditions resulted in 36,729, 60,352, and 77,926 ATAC-seq peaks for *S. polychroa*, *S. nova*, and *S. lugubris*, respectively (Supplementary Data 7-9). To assess the conservation of *S. mediterranea* ATAC-seq peaks relative to these data, we assessed sequence conservation under each peak through whole genome alignment liftover, and chromatin state conservation by overlapping the whole genome alignment liftover with ATAC-seq data in the receiving species (Fig. 3f). Peaks from *S. mediterranea* with only sequence conservation in the receiving species were designated as conserved regions; conserved regions overlapped with an ATAC-seq peak in the receiving species were designated as conserved peaks. This allowed us to categorize each *S. mediterranea* peak according to its conservation status across the genus, with the most informative categories being 'not conserved', 'partial conservation' (all categories except not conserved), and 'high conservation' (3 peaks, Fig. 3g, Supplementary Data 1) in the case of peaks that showed conservation in all three recipient species. Across

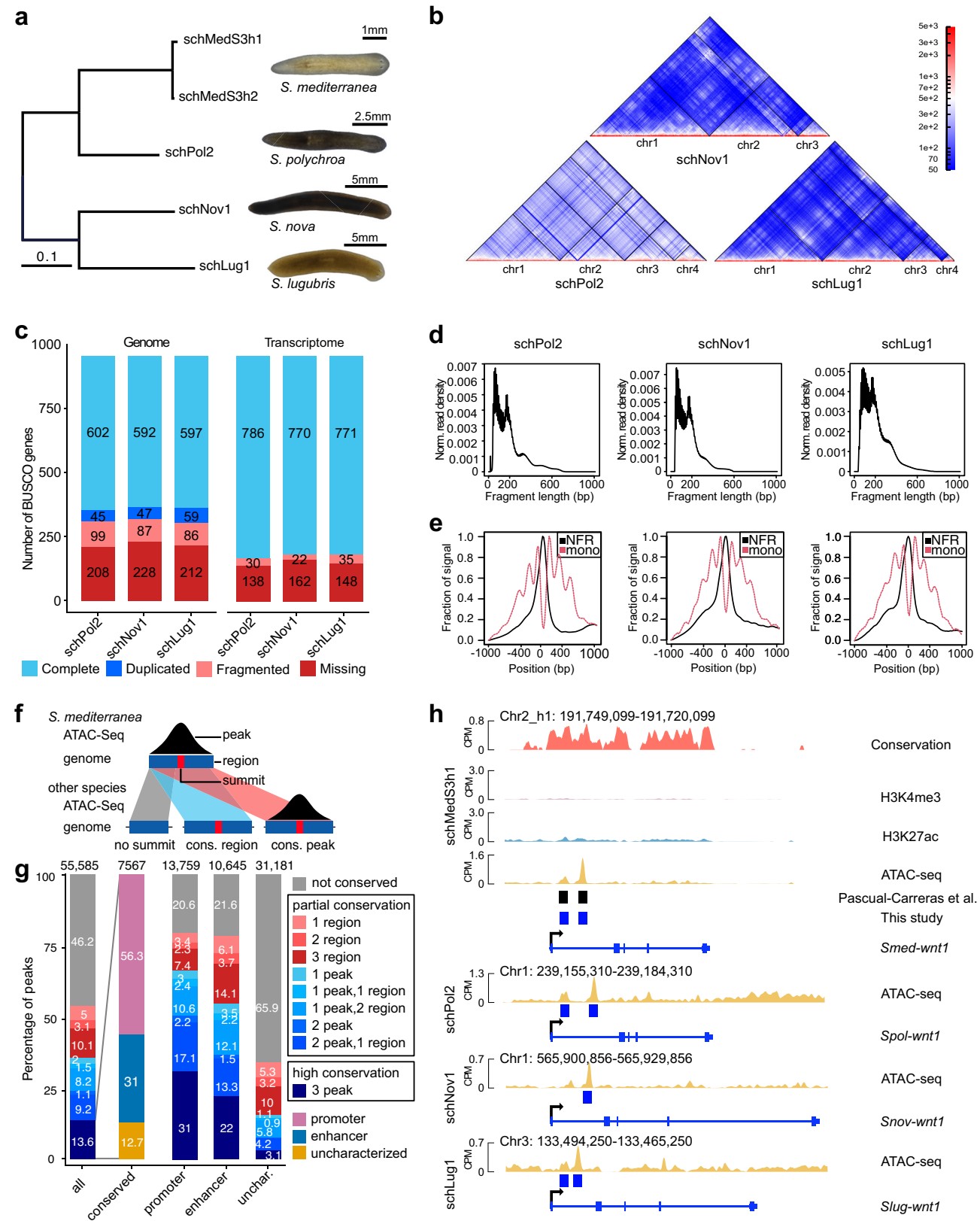

all 55,585 ATAC-seq peaks annotated in *S. mediterranea*, 13.6% were highly conserved, 40.2% were partially conserved, and 46.2% were not conserved (Fig. 3g, Supplementary Data 10–12). Interestingly, 87.3% of the highly conserved ATAC-seq peaks also had additional ChIP-seq support (Fig. 3g). Furthermore, when considering all ATAC-seq peaks with additional ChIP-seq support, we found that 79.4% of the putative

promoters and 78.4% of the putative enhancers displayed at least partial conservation, thus confirming that these peak sets are indeed likely to be enriched for functionally relevant regions under purifying selection (Supporting Information: Section 3). As expected, the 30,197 "uncharacterized" ATAC-seq peaks were collectively much less conserved, with only 3.1% of high conservation and 34.1% of partial

**Fig. 3 | Comparative ATAC-seq in the genus *Schmidtea* to assess regulatory element conservation. a** Evolutionary distance of the analyzed *Schmidtea* species on the basis of 4-fold degenerate sites in the whole genome alignments. Branch lengths indicate expected substitutions per site. The tree topology is based on a phylogenomic analyses of 930 single-copy genes and agrees with previous work[59]. **b** Hi-C contact maps of the genome assemblies of *S. polychroa* (schPol2), *S. nova* (schNov1), and *S. lugubris* (schLug1). **c** BUSCO completeness assessment of the genome assemblies (left three bars) and the corresponding annotations (right bars) of the new *Schmidtea* genomes. Since the annotation assessment was run on the transcript level, the "Duplicate" and "Complete" category were combined for the visualization to avoid apparent duplication due to isoforms. **d** ATAC-seq fragment-size distributions of the indicated species. **e** Pileups of nucleosome-free region (NFR) and mononucleosomal (mono) reads at transcription start sites (TSS) of the indicated species. **f** Schematic diagram showing the definition of a conserved region (sequence conservation without ATAC-seq signal) and conserved peak (sequence conservation and ATAC-seq signal). **g** Bar plots indicate the conservation status of all *S. mediterranea* ATAC-seq peaks by conservation category and the degree of conservation across the sister species. The "conserved" bar sub-categorizes the 13.6% of highly conserved peaks according to the histone mark categorization in Fig. 2i. The three bar plots represent the conservation status of putative promoters, putative enhancers, and uncharacterized ATAC-seq peaks. **h** Highly conserved regulatory elements in association with the *wnt1* gene locus. Top: Representation of the *S.mediterranea* gene locus, including sequence conservation (PhyloP on whole genome alignments), H3K4me3 ChIP, H3K27ac ChIP, and ATAC-seq tracks, annotated regulatory elements (published[39], black; this study, blue), and the exon/intron representation of the *wnt1* gene. Below, the available tracks in the three sister species genomes are shown aligned by the TSS. Note, that only a single peak was called in *S. nova* and the peak was therefore classified as a conserved region in that species. Source data are provided as a Source Data file.

conservation (Chi-squared = 9745.1, df = 4, *p*-value < 2.2e-16, all post-hoc tests *p* < 0.001, Supporting Information: Section 3). Thus, although the partially conserved subset may contain functionally relevant sequences, the nature and functional significance of most *Schmidtea*-specific ATAC-seq peaks remain unclear. Consistent with the considerable sequence divergence between the four *Schmidtea* species, the ~20% of ATAC-seq peaks that are *S. mediterranea*-specific but ChIP-annotated further indicate a likely significant degree of regulatory divergence within the genus.

Owing to the comparatively recent advent of functional genomics in the field, only two gene-enhancer sequences have been partially characterized[39,61]. To gauge the practical utility of our regulatory element analysis, we first examined the putative *wnt1* enhancers of *S. mediterranea*[39]. As shown in Fig. 3h, we also identify two ATAC-seq peaks in the first intron, which we categorize as putative intronic enhancers due to their overlap with the H3K27ac signal. In addition, our conservation analysis identifies one region as "highly conserved", with prominent ATAC-seq peaks in all species, and one region as having a conserved peak in *S. polychroa* and *S. lugubris* but only a conserved region in *S. nova* (Fig. 3h). Interestingly, the prominence of the ATAC-seq peaks in all four species and the H3K27ac peak in *S. mediterranea* contrast with low H3K4me3 signals at the *S. mediterranea* TSS (Fig. 3h, top). While the latter is consistent with the highly specific expression of *wnt1* in very few cells at the tail tip of intact animals, the former might indicate that the regulatory regions of *wnt1* are constitutively accessible in a much broader range of cells to allow its dramatic upregulation at any *S. mediterranea* wound site[62]. In contrast, our analysis did not detect the proposed head-specific enhancer sequence of *nou-darake* (*ndk*)[62], even though multiple other putative regulatory sequences near the gene were annotated (Supporting Information: Section 3). While this may well reflect the limitations of our current whole-animal datasets in detecting regulatory regions that are only active in a small number of cells, our summary of both the ChIP-seq and conservation status of all *S. mediterranea* ATAC-seq peaks (Supplementary Data 1) nevertheless provides a valuable resource for reconstructing regulatory circuits in the model species and their evolutionary divergence across the genus *Schmidtea*.

## Genome architecture & synteny

The availability of the four chromosome-scale genome assemblies also provided a first opportunity for exploring other features of genome evolution within the taxon. As noted, *S. lugubris* and *S. nova* had substantially larger genomes than *S. polychroa* and *S. mediterranea* (Fig. 4a). Transposable element annotations revealed that a large proportion of the increase in genome size can be attributed to an expansion of transposable elements, in particular, DNA and LTR/Gypsy elements (Fig. 4a). Furthermore, in *S. lugubris* and *S. nova*, the total gene span was 54% and 28% larger compared to *S. mediterranea*. This increase was primarily due to the increased length of protein-coding genes and, specifically, the expansion of introns, at least in parts due to transposon insertions (Table 1, Supplementary Data 4). Therefore, the larger assembly sizes of *S. lugubris* and *S. nova* reflect genuine genome size expansions due to transposable element expansions and the identification of the expanded transposon families is an interesting topic for future investigations.

Next, we assessed the synteny between the four genomes using GENESPACE[63]. The visualization of the syntenic blocks revealed a large number of rearrangements between the genomes (Fig. 4b). Already between the two pseudo-haplotypes of *S. mediterranea*, the previously noted large inversion on Chromosome 1 and the smaller inversion on Chromosome 2 stand out as prominent structural rearrangements (indicated with red bars and dark shading in Fig. 4b). Comparisons with the other genomes revealed a striking history of frequent structural rearrangements encompassing inversions and inter-chromosomal translocations. For instance, the gene content of *S. mediterranea*'s Chromosome 1 is split between *S. polychroa*'s chromosomes 3 and 4, and *S. polychroa*'s Chromosome 2 is equivalent to *S. mediterranea*'s chromosomes 3 and 4, suggesting splits and fusions between all chromosomes (Fig. 4b). Moreover, the chromosomal reduction in *S. nova* from 4 to 3 implies a complex fusion event between multiple chromosomes. The rapidly decreasing size and increasing number of syntenic blocks in the assembly comparisons quantitatively confirmed the chromosomal fragmentation apparent in Fig. 4b (See Supporting Information: Section 4.1 for more details and Supplementary Data 13, 14 for the inferred syntenic blocks). Interestingly, we found that 10 kb windows flanking the synteny breakpoints in our species panel were significantly enriched in LTR/Gypsy retrotransposons in all assemblies except for *S. nova* and enriched in LINE/R2 retrotransposons in all assemblies except *S. nova* and *S. lugubris* (Fig. 4c, d), thus providing a first indication that transposable elements may play a role in the frequent structural rearrangements as is the case for LINE elements in humans[64] (Supporting Information: Section 4.2 and Supplementary Data 15). Overall, our synteny analysis revealed a surprising amount of structural genome rearrangements within the genus *Schmidtea*, including frequent inter-chromosomal translocations and the consequent erosion of gene order within chromosomes.

Intrigued by these findings, we next asked whether this feature is unique to *Schmidtea* or if synteny is generally poorly conserved amongst flatworms. To address this, we selected chromosome-scale genome assemblies of four parasitic flatworms (Neodermata), comprising the two Trematoda species *Clonorchis sinensis* and *Schistosoma mansoni*, and two Cestoda species *Taenia multiceps* and *Hymenolepis microstoma*. We collectively refer to these as "parasites" in the following. Additionally, we included a highly contiguous – but not chromosome-scale – genome assembly of *Macrostomum hystrix* in some subsequent analyses to represent the early branching flatworm

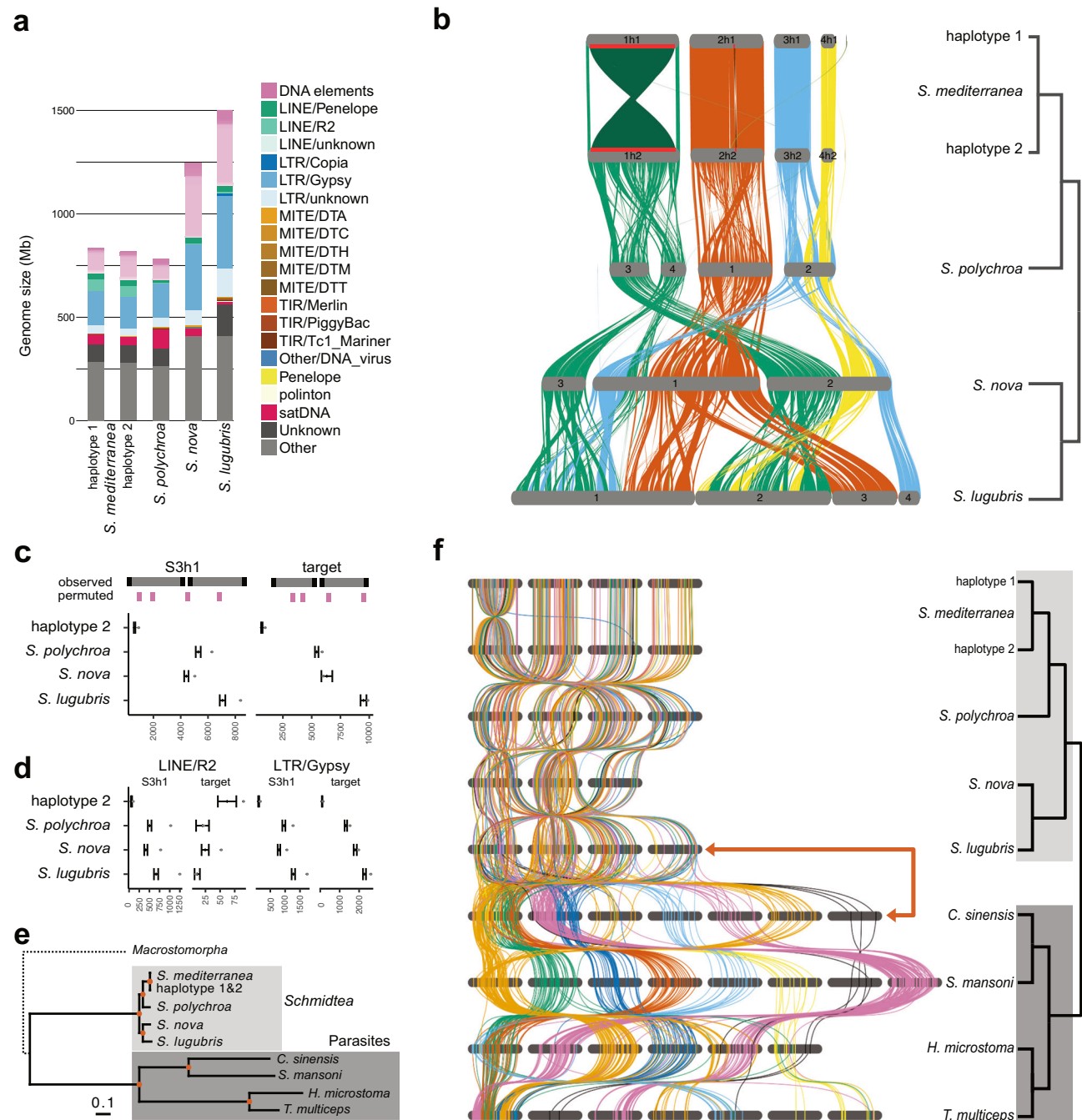

**Fig. 4 | Structural evolution of flatworm genomes. a** Bar plot of the sizes and repetitive element contributions to the indicated *Schmidtea* assemblies. **b** Synteny analysis between the four *Schmidtea* species. Ribbon coloring on the basis of *S. mediterranea* chromosome locations. Red bars and darker shading in schMedS3h1 indicate the inversions on Chromosome 1 and 2 that distinguish haplotype 1 and 2. **c, d** Enrichment analysis of 10 kb windows flanking synteny breakpoints, inferred using GENESPACE, in the *Schmidtea* assemblies. Tests compare the observed value (black dots) to 1000 random permutations of 10 kb windows in the reference (colored mean and standard deviations). Black dots outside the permuted range indicate statistically significant enrichment. Shown are the results for all transposable elements (c) and LINE/R2 and LTR/Gypsy elements (d). See Supporting Information: Section 4.2 and Supplementary Data 15 for details. **e** Phylogenetic relationship of *Schmidtea* and the indicated parasitic flatworm species (parasites). The maximum-likelihood phylogeny is based on a concatenated alignment of 930 single-copy orthologs with a total of 818,016 aligned amino-acid positions. Red dots on nodes indicate maximum ultra-fast bootstrap and SH-like approximate likelihood ratio test support. The phylogeny was rooted at its midpoint. The early-branching Macrostomorpha are indicated with a dotted line. **f** Synteny analysis based on BUSCO gene positions between the indicated *Schmidtea* and parasitic flatworm assemblies. Genes are represented with bars colored based on the chromosome location in *Schistosoma mansoni*. Source data are provided as a Source Data file.

group Macrostomorpha. As expected from the large evolutionary distances involved, the protein-sequence divergence of the parasite genomes was much higher than the divergence within the *Schmidtea* taxon (Fig. 4e). To obtain an overview of synteny conservation across such evolutionary distances, we first compared the chromosome-scale

assemblies using the location of single-copy genes based on de novo BUSCO annotations. As shown in Fig. 4f, large genomic blocks of BUSCOs (color blocks; parallel lines) were conserved across the para-sites despite several large-scale genomic rearrangements. Intriguingly, between the parasites and the *Schmidtea* genomes (red arrow), the

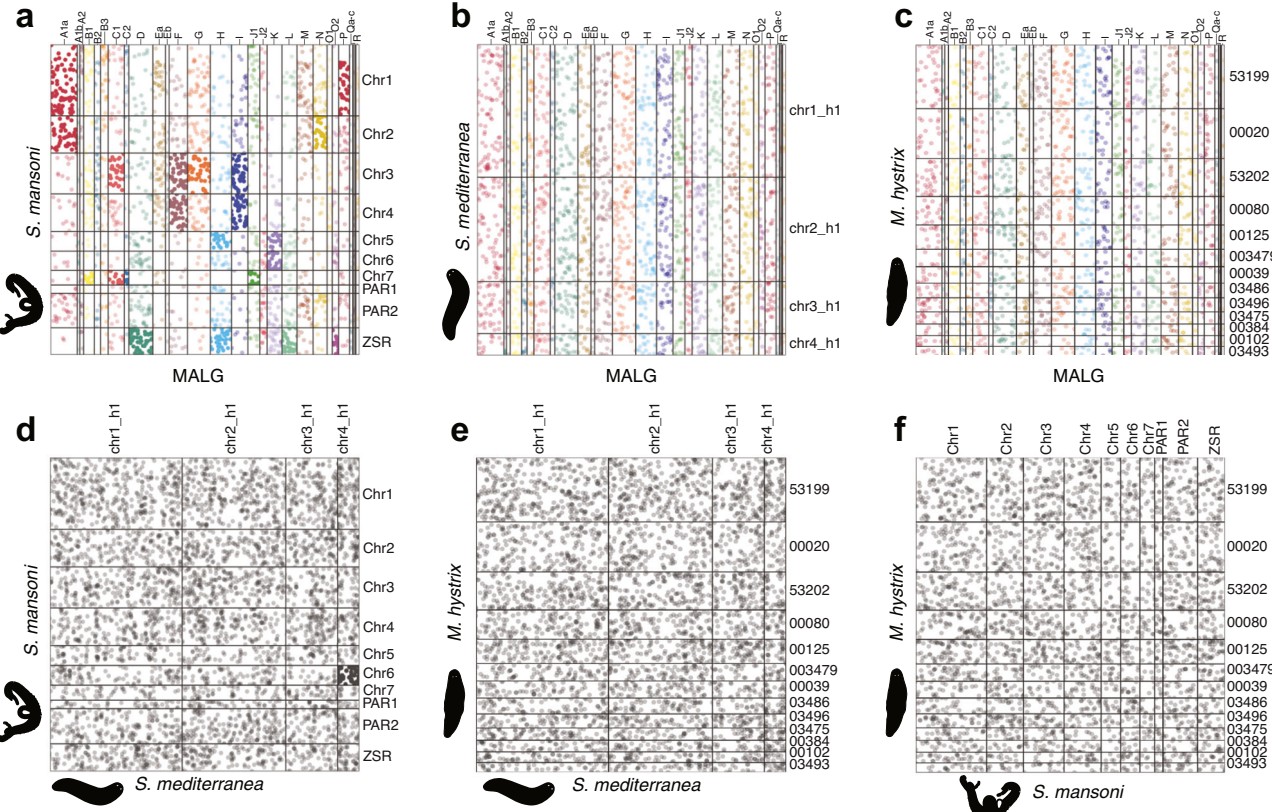

**Fig. 5 | Lack of metazoan ancestral linkage group (MALG) and synteny conservation in flatworms.** Dotplots between metazoan ancestral linkage groups (MALG) and **a** *Schistosoma mansoni*, **b** *Schmidtea mediterranea*, and **c** *Macrostomum hystrix*. MALGs that are significantly enriched in one or more chromosomes according to a Fisher Exact test are indicated in dark color, while MALGs without enrichment are plotted in light colors. No enrichment was detected for any MALG in *S. mediterranea* and *M. hystrix*. **d**–**f** Synteny analysis between *Schistosoma mansoni*, *Schmidtea mediterranea*, and *Macrostomum hystrix*. Boxes indicate chromosome combinations and dots represent one-to-one orthologs inferred with ODP. No chromosome combination was enriched except for 57 orthologs between *S. mansoni* and *S. mediterranea* located on Chromosome 6 and Chromosome 4, respectively, indicating that synteny between these three flatworm groups is not conserved. The outline of *S. mansoni* was drawn with permission based on artwork by Guido Hegasy. Source data are provided as a Source Data file.

relative positions of the same BUSCO genes appeared largely randomized. We quantitatively confirmed this finding using a 9-fold expanded gene set using Orthofinder and Chi-squared test analyses, which again revealed the maintenance of measurable synteny within the *Schmidteas* and the parasites, but the near-total degradation of synteny between the two groups (e.g., small residual effect sizes with all < 0.09, and with the Chi-square test not even reaching the significance threshold for the schMedS3h1-hymMic and schMan-schNov1 combinations; Supporting Information: Section 5.3). The analysis of chromosome gene complement conservation between *Schmidtea*, the parasites, and the early-branching flatworm *Macrostomum hystrix* using 1:1 orthologue annotations inferred using the ODP tool (see Methods) also revealed the near complete absence of synteny between the two groups (Supporting Information: Section 5.4). Overall, this analysis confirmed a profound loss of synteny and chromosomal gene content between the genus *Schmidtea*, parasitic flatworms, and the early-branching flatworm *Macrostomum hystrix*.

This result also raised the question of which flatworm groups better conformed to the 28 ancestral metazoan linkage groups (hereafter MALGs) identified by Simakov et al.[8] We, therefore, identified orthologs of the MALGs using the ODP tool to determine if they were associated with particular chromosomes of our test genomes. Half of the 28 MALGs were statistically significantly enriched on specific chromosomes in *Schistosoma mansoni* (Fig. 5a, Supporting Information: Section 4.5). However, all linkage groups were either spread across several chromosomes and/or fused and mixed with other linkage groups (see our detailed description of these mixing events in Supporting Information: Section 4.5). A similar pattern was observed in the other parasites, with 12, 6, and 12 MALGs partially conserved in *Clonorchis sinensis*, *Hymenolepis microstoma*, and *Taenia multiceps*, respectively. Notably, despite the conservation of synteny between the parasites, the statistical tests did not find conservation of the same MALGs in each species (Supporting Information: Section 4.5). This suggests that the parasites retain detectable traces of the MALG, although these have been largely obscured by a history of widespread chromosomal rearrangements. In sharp contrast, we were unable to detect any traces of MALG conservation in the *Schmidtea* genomes (Fig. 5b, Supporting Information: Section 4.5) and the *Macrostomum hystrix* assembly (Fig. 5c, Supporting Information: Section 4.5). Remarkably, our ODP analysis further revealed the lack of any detectable synteny conservation between *Macrostomum*, *Schmidtea*, and the parasites, therefore suggesting that these taxa represent entirely independent genome architectures (Fig. 5d–f, Supporting Information: Section 4.5). Overall, the extensive gene shuffling between the genomes analyzed indicates that synteny may not be constrained in flatworm evolution.

## Discussion

Here, we present and analyze four genomes of planarian flatworms in the genus *Schmidtea*, including a chromosome-scale and haplotype-phased assembly of the model species *S. mediterranea*. The new *S. mediterranea* assembly is more contiguous, complete, and has a higher

sequence accuracy than our previous assembly[26] (Fig. 1). We further provide new gene annotations that improve upon the current community standards (dd_v6 transcriptome, SMESGD annotations). Complementing similar recent efforts[38–40], we annotated regulatory regions in the *S. mediterranea* genome using ATAC-seq and ChIP-seq and integrated evolutionary sequence conservation across *Schmidtea* genomes as orthogonal evidence to enrich for functional regulatory elements. Overall, our study provides valuable resources for reconstructing gene regulatory circuits in *S. mediterranea* and the planarian research community in general. Nevertheless, the current S3 *S. mediterranea* genome assembly is not yet finished. Remaining challenges include the fine scaffolding of the currently ~400 unscaffolded contigs containing > 500 annotated genes or a low percentage of annotation errors in the current gene annotation (e.g., the erroneous fusion between the Activin inhibitor *follistatin* to the gene immediately upstream, see Supporting Information: Section 1.5). Therefore, the adequate versioning of future assemblies and gene annotations and the incorporation of manual curation into PlanMine[42] and other resources (e.g.[65–68]) represent strategic objectives for the planarian research community.

Beyond *S. mediterranea* resources, our study's four high-quality *Schmidtea* genomes present a first opportunity to explore patterns of genome evolution within the genus and in planarians in general. With third base sequence divergences equivalent to ~30 Mio years (*S. mediterranea* vs. *S. polychroa*) or ~70 Mio years (*S. mediterranea* vs. *S. nova*) of vertebrate genome evolution[60] amongst the closest known sister species of *S. mediterranea*, the four genomes further emphasize the extent of sequence divergence within planarians[41]. In addition, the assemblies reveal a striking degree of structural divergence between the *Schmidtea* genomes. As already apparent between the two haplotypes of the S3 assembly (Fig. 1e), large-scale structural rearrangements dominate the *Schmidtea* species genome comparisons. The unbalanced chromosomal translocation between the sexual and asexual biotypes of *S. mediterranea*[69] and previous taxonomic classification of *Schmidtea* into biotypes based on karyotypes[59] have already hinted at frequent structural rearrangements in *Schmidtea* species. Our chromosome-scale assemblies now confirm the hypothesis that Chromosome 1 of *S. nova* resulted from a fusion of chromosomes 1 and 3 of *S. lugubris*[58] and additionally reveal that *S. lugubris* Chromosome 1 also shares synteny with Chromosome 3 of *S. nova*. Similarly, we find that the *S. mediterranea* Chromosome 1 synteny block was split into two parts in the other three species, suggesting its origin in a fusion with mixing event. These multiple 'hidden' rearrangements are similar to what has recently been uncovered in holocentric Lepidoptera and beaksedges[70,71]. However, unlike in the Lepidoptera and consistent with results in beaksedges[71], we discovered an enrichment of the abundant LTR/Gypsy elements and LINE/R2 elements near the synteny breakpoints, which may hint at the underlying mechanisms for the rearrangements (see below). The consequence of the large scale and high frequency of structural rearrangements in the genomes is a striking degradation of synteny across the *Schmidtea* genomes.

Our results further suggest that such structural genome instability is not specific to *Schmidtea* but a general feature of flatworm genome evolution. The striking qualitative randomization of BUSCO genomic positions (Fig. 4e) and the quantitative analysis approach of Metazoan ancestral linkage groups (MALG; Fig. 5a–c[8]) demonstrate the near-complete absence of genomic synteny between *Schmidtea*, the analyzed parasite genomes and even the genome of the early-branching flatworm *Macrostomum hystrix*. These results imply that the genome architectures of the sampled flatworm clades have evolved largely independently, which also raises a cautionary note regarding the interpretation of the previously suggested selective enrichment of the *S. mansoni* sex chromosome genes on Chromosome 1 of *S. mediterranea*[27]. Moreover, our finding that MALGs have been lost independently in *Schmidtea* and *M. hystrix* but weakly retained in the

parasites (also see ref. 72) is remarkable, given that they are often assumed to represent the most derived taxonomic group[73] based on their phylogenetic position[52,53], compacted genomes[72], and obligate parasitic life cycles[74]. A broader taxon sampling of flatworm genomes will be required to place the evolution of parasitism within the evolutionary history of the phylum. In addition, the loss of MALGs per se is uncommon, given that MALGs are defined based on their conservation across metazoan genomes[8,9,75,76], and some are even conserved in unicellular relatives of animals[10]. Although MALG losses have already been noted in other taxonomic groups[11,12,77], the obliteration of ancestral linkage groups at the base of these lineages appears to have been followed by the establishment and retention of new clade-specific linkage groups (e.g., Nigon elements in Nematodes[12], Muller elements in Drosophilids[11], ALGs in Bryozoa[77]). Interestingly, the group-specific linkage groups remain even in clades that contain species with drastic genome/karyotype rearrangements, suggesting selection for the maintenance of linkage groups rather than mechanistic constraints on inter-chromosomal rearrangements[10]. In contrast, our results indicate that the dispersal of MALGs in flatworms was not accompanied by the emergence of clade-specific linkage groups and that gene order has been and continues to evolve independently within the different taxa. Altogether, this amounts to the provocative proposition that synteny may not matter in flatworms.

One of the interesting questions raised by our findings is how flatworms can achieve gene expression specificity, apparently without the topological constraints that are important in other systems[15–21]. An indication that planarians may achieve gene expression specificity by unusual means is that topologically associated domains (TADs) are not apparent in our Hi-C data (Figs. 1a, 3b), and although A/B compartments can be called in *S. mediterranea*, they do not show sharp boundaries or the typical partition into gene-rich and gene-poor compartments[40]. Thus, planarians appear more similar to cnidarians[76], which lack TADs or *C. elegans*, where TADs on the autosomes are less pronounced[78] and classic TADs are restricted to the X chromosome[79]. In addition, the large fraction of intronic enhancers and the fact that only 20% of putative enhancers were classified as "distal intergenic" raises the possibility that regulatory elements in *S. mediterranea* may be tightly associated with genes. Such a tight association between regulatory elements and genes has also been proposed in a tunicate species complex with rapid but chromosome-arm-restricted gene shuffling[80]. In addition, the recent finding that many planarian genes may be regulated by an interplay between AT-rich motifs in the vicinity of the TSS via nucleosome positioning[40] may further contribute to a gene-centric regulatory structure. However, our current association of genes with putative enhancers is based solely on proximity, and systematic functional analyses will be required to demonstrate that planarian genes are indeed more tightly associated with their regulatory elements than in other species.

A further interesting question raised by our study are the mechanistic causes of the frequent genome rearrangements in *Schmidtea*. Both free-living and parasitic flatworms survive gamma-irradiation well beyond lethal doses in vertebrates[81–84], which implies the existence of efficient double-strand break repair pathways that are also known to mediate Robertsonian translocations[85–87]. The enrichment of LTR/Gypsy and LINE/R2 elements near the synteny breakpoints might indicate the role of retrotransposons as templates for strand invasion during the repair of double-strand breaks. Whether double-strand break repair pathways mediate chromosomal rearrangements and ultimately drive the structural evolution of planarian genomes is, therefore, a further interesting topic for future analysis. Finally, the possibility that parallel somatic evolution and selection phenomena amongst a single planarian's many thousand pluripotent somatic stem cells might contribute to the extraordinary rates of sequence divergence raises profound questions regarding the maintenance of genetic "self" and the evolution of multicellularity[88]. In

summary, understanding the mechanistic links between the unusual patterns of genome evolution of flatworms and their unusual biology remains a fascinating research endeavor.

## Methods

### Samples

All animals used for these analyses were derived from long-term laboratory cultures maintained at the Max Planck Institute of Molecular Cell Biology and Genetics in Dresden and the Max Planck Institute for Multidisciplinary Sciences in Göttingen. The animals were kept in planarian water supplemented with gentamycin sulfate at 50 µg/mL at 20 °C and fed with organic calve liver as described previously[89]. We used the laboratory strain of the sexual biotype of *S. mediterranea* originating from Sardinia that was also used for the previous genome project (S2F18, derived from S2F2, internal ID: GOE00500). Functional data was generated from the standard laboratory strain of the asexual biotype of *S. mediterranea* (CIW4, internal ID: GOE00071). The *S. nova* strain (internal ID: GOE00023) was collected at 51,0717710 and 13,7421400 in Dresden, Germany, on 2013-04-14. The *S. lugubris* strain (internal ID: GOE00057) was collected at 52.942432, −1.113739 in Nottingham, UK (JCR). The S. *polychroa* strain (internal ID: GOE00227) was collected at 43.71249; 16.72605 near the Village of Gala, Croatia. Animals were starved for ten days prior to experiments. For the x-ray irradiation treatment, animals were irradiated with 60 Gray using a Precision Cellrad Cell Irradiation System (10-130 KV, Precision X-Ray, USA).

### High molecular weight DNA extraction

High molecular weight DNA was extracted as previously described with modifications[26]. Briefly, planarians were treated with a 0.5% (w/v) N-acetyl-L-cysteine (NAC) stripping solution, augmented with 20 mM HEPES-NaOH at a pH of 7.25. The pH was carefully adjusted to approximately 7 using 1 M NaOH and monitored using a 0.5% (w/v) phenol-red solution. Planarians were submerged in 10 mL of this freshly prepared NAC solution and agitated vigorously, for instance, on a rotator, for 10 minutes at room temperature. After this, a quick rinse with distilled water was done before proceeding to the DNA extraction phase.

Only wide-bore pipette tips were utilized for DNA isolation to handle the high molecular weight DNA, ensuring minimal shear forces. About 20 mucus-stripped planarians, roughly 1 cm in size and starved for 1-3 weeks, were placed into a 50 mL tube. They were then lysed using 15 mL of cold GTC buffer (containing 4 M guanidinium thiocyanate, 25 mM sodium citrate, 0.5% (w/v) N-Lauroylsarcosine, and 7% (v/v) β-mercaptoethanol) for 30 minutes on ice, with the tube being inverted every 10 minutes to promote tissue dissociation. The lysate was mixed with an equal volume of phenol/chloroform/isoamyl alcohol (in a 25:24:1 ratio), buffered with 10 mM Tris pH 8.0 and 1 mM EDTA. This mixture was centrifuged at 4000 x *g* for 20 minutes at 4 °C, after which the upper aqueous phase was carefully collected into a new tube. The phenol/chloroform extraction step was repeated 1-2 times, or until the interphase vanished. Any remaining phenol was removed by mixing the aqueous phase once with an equivalent volume of chloroform, followed by centrifugation. To this cleared aqueous phase, an equal volume of ice-cold 5 M NaCl was added and mixed. After a 15-minute incubation on ice, the sample was centrifuged at maximum speed for 10 minutes at 4 °C to pellet any contaminants. The nucleic acid-rich supernatant was moved to a new tube, precipitated using 0.7-1 volumes of isopropanol, and centrifuged at 2000 x *g* for 30–45 minutes at room temperature. The DNA pellet was then washed with 70% ethanol, centrifuged at 2000 x *g* for 5 minutes, briefly air-dried, and finally resuspended in 50 µL TE buffer and left to dissolve overnight at 4 °C.

During post-purification with CTAB at room temperature, contaminants were removed from the isolated DNA. The DNA was treated with 1 µL RNase A (4 mg/mL) for an hour at 37 °C, and NaCl concentration was adjusted using a 2% CTAB/1.4 M NaCl solution. After mixing with chloroform and centrifuging at 12,000–16,000 x *g* for 15 minutes, the clear phase was extracted. The DNA was then precipitated using isopropanol, washed in 70% ethanol, and resuspended in TE buffer overnight at 4 °C.

It is known that DNA can be removed from crude lysates by streptomycin complexation[90,91]. Compared to other aminoglycoside antibiotics, streptomycin has negligible affinity for acidic mucopolysaccharides, proteins, and RNA[92]. We therefore employed streptomycin precipitation to specifically separate DNA from remaining contaminants. CTAB-purified DNA samples in a 1.5 mL low DNA binding tube (Eppendorf) were mixed with 0.1–0.2 volumes of 50 mg/mL streptomycin sulfate in nuclease-free H2O. For highly viscous samples, the sample was diluted by additional TE buffer before streptomycin addition. The mixture was carefully shaken or flicked to avoid DNA shearing. The DNA-streptomycin complex was allowed to form for at least 15 minutes at room temperature and was subsequently precipitated by centrifugation at 4000 x *g* for 30 minutes at room temperature. The supernatant was removed without disturbing the pellet. Excess streptomycin was washed off using 1 mL of PEG/NaCl-based wash buffer (10% (w/v) PEG-8000, 1.25 M NaCl, 10 mM Tris-HCl pH 8.0, 1 mM Na2EDTA pH 8.0, 0.05% (v/v) Tween-20) for 15 minutes at room temperature which keeps the DNA precipitated[93]. After centrifugation for 5 minutes at 4000 x *g* at room temperature, the supernatant was removed, the pellet briefly washed with 1 mL of 70% ethanol, and pelleted as before. After removal of 70% ethanol the pellet was resuspended in 100 µL of DNA pre-dialysis buffer (10 mM Tris-HCl pH 9.0, 2 M NaCl, 1 mM Na2EDTA, pH 9.0). A dialysis membrane (Millipore: VSWP 04700 (mean pore size = 0.1 µm)) was hydrated on a 100-fold volume of DNA dialysis buffer (10 mM Tris-HCl, pH 9.0, 0.1 mM Na2EDTA, pH 9.0). The sample was carefully transferred onto the dialysis membrane and dialyzed for 4-6 h at RT and then carefully collected using a wide-bore pipette tip. The quality and quantity of the DNA were verified using pulse field gel electrophoresis run using the Pippin Pulse™ device (SAGE Science), and the Qubit™ fluorometer.

### PacBio High Fidelity (HiFi) library preparation and sequencing

For all species the genomic DNA entered library preparation using the PacBio HiFi library preparation protocol "Preparing HiFi Libraries from Low DNA Input Using SMRTbell Express Template Prep Kit 2.0". Briefly, all gDNA was sheared to 14-22 kb with the MegaRuptor device (Diagenode) and 12–18 µg sheared gDNA was used for library preparation. Depending on the gDNA input amount and performance during library preparation, the PacBio libraries were either size selected for fragments either larger than 3 kb with Ampure beads or for fragments larger than 8-10 kb using the BluePippin™ device. The size-selected libraries were prepared for loading following the instructions generated by the SMRT Link software (PacBio, version 10) and the 'HiFi Reads' application. The Sequel® II Binding Kit 2.2 (PacBio, USA) was used to prepare the libraries for loading, using the Sequel® II DNA Internal Control Complex 1.0 (PacBio). All libraries ran on SMRT™ Cells 8 M (PacBio) using the Sequel® II Sequencing Kit 2.0 (PacBio) on the Sequel® II Sequencer (PacBio).

### Phased genome assembly of *S. mediterranea*

Circular consensus sequences from ~30x coverage PacBio reads were called using pbccs (v6.0.0, https://github.com/nlhepler/pbccs) and reads with quality > 0.99 (Q20) were taken forward as "HiFi" reads. Additionally, we generated 1000 million Hi-C reads from extracted nuclei of whole animals using the Arima-HiC+ Kit. PacBio HiFi and Hi-C reads were used to assemble phased contigs with hifiasm (v0.7,[94]). Next, Hi-C reads whose mapping quality no less than 10 (-q 10) were further utilized to scaffold the contigs from each haplotype by SALSA (v2,[95]) following the hic-pipeline (https://github.com/esrice/hic-pipeline),

which includes filtering procedures such as removal of experimental artifacts from Hi-C alignments, fixing of Hi-C pair mates, and removal of PCR duplicates, etc. Four chromosome-level scaffolds could be observed in both haplotypes after scaffolding. However, Hi-C heatmap revealed evidence of misplacement of contigs in terms of positions and orientations. These errors were then manually curated based on the interaction frequency indicated by the intensity of Hi-C signals. To compare the haplotypes with each other and compare them to the schMedS2 assembly, we aligned them using minimap2 (-asm5, v2.24,[96]) and parsed the alignments using syri (v1.6.3,[97]).

### Genome assembly of *S. polychroa*, *S. nova*, and *S. lugubris*

Circular consensus sequences from PacBio reads were called using pbccs (v6.0.0) and reads with quality > 0.99 (Q20) were taken forward as "HiFi" reads. To create the initial contig assemblies for *S. nova*, canu v2.1 was used with parameters: maxInputCoverage=100 -pacbio-hifi. For *S. polychroa* and *S. lugubris* hifiasm (v0.14.2) was used to create initial contigs with purging parameter: -l 2. Next, alternative haplotigs were then removed using purge-dups (v1.2.3) using default parameters and cutoff as they were correctly estimated by the program. To initially scaffold the contigs into scaffolds, SALSA v2 (v2.2) was used after mapping Hi-C reads to the contigs. The VGP Arima mapping pipeline was followed: https://github.com/VGP/vgp-assembly/tree/master/pipeline/salsa using bwa-mem (v0.7.17), samtools (v0.10, v1.11) and Picard (v2.22.6). False joins in the scaffolds were then broken and missed joins merged manually following the processing of Hi-C reads with pairtools (v0.3.0) and visualization matrices created with cooler (v0.8.11).

Following scaffolding, the original PacBio subreads were mapped to the chromosomes using pbmm2 (v1.3.0, https://github.com/PacificBiosciences/pbmm2) with arguments: --preset SUBREAD -N 1 and regions +/- 2 kb around each gap were polished using gcpp's arrow algorithm (v1.9.0). Those regions in which gaps were closed and polished with all capital nucleotides (gcpp's internal high confidence threshold) were then inserted into the assemblies as closed gaps.

Lastly, the PacBio HiFi (CCS reads with a read quality exceeding 0.99) were aligned to the genomes using pbmm2 (v1.3.0) with the arguments --preset CCS -N 1. DeepVariant (v1.2.0,[98]) was used to detect variants in the alignments to the assembled sequence. Only the homozygous variants (GT = 1/1) that passed DeepVariant's internal filter (FILTER = PASS) were retained using bcftools view (v1.12) and htslib (v1.11). The genome was then polished by creating a consensus sequence based on this filtered VCF file, as detailed in the VGP assembly pipeline (https://github.com/VGP/vgp-assembly/tree/master/pipeline/freebayes-polish).

### Bacterial and mitochondrial sequence removal

We used FCS-GX (https://github.com/ncbi/fcs) to screen the genome assemblies for any potential bacterial and fungal sequences. Contigs that were flagged with 'EXCLUDE' because they contained a fungal or bacterial hit were removed. Based on our findings we removed 42 contigs from the schMedS3h1, 12 contigs from the schMedS3h2, 5 contigs from the schPol2, and 7 contigs from the schLug1 assembly. Furthermore, we removed one contig from the schMedS3h1 assembly because it represented the mitochondrial genome.

### De novo repeat discovery and annotation

We annotated transposable elements using the Extensive de novo TE Annotator (EDTA) workflow (v2.1.0,[99]). This approach augments the standard Repeat Modeler workflow with additional tools specifically targeted at LTR, Helitron, and TIR-Elements. We used parameters: '--species others --step all --sensitive 0 -anno 1' and provided the previously manually curated repeat library generated for *S. mediterranea*[26] as a curated library. Additionally, Transposable element protein domains (Neumann et al., 2019) found in the assembled genomes were

annotated using the DANTE tool available from the RepeatExplorer2 Galaxy portal (https://repeatexplorer-elixir.cerit-sc.cz/galaxy/) exploiting the REXdb database[100] (Viridiplantae_version_3.0).

To identify the overall repetitiveness of the genomes we performed de novo repeat discovery with RepeatExplorer2[101]. For *S. mediterranea* we used a repeat library obtained from the RepeatExplorer2 analysis of shotgun whole-genome Illumina paired-end sequencing (NCBI accession: SRR5408395). Since for *S. polychroa*, *S. nova*, and *S. lugubris* no Illumina data was available, we generated pseudo paired-end reads from 2 Gb of CCS reads as input for RepeatExplorer2. All clusters representing at least 0.005% of the genomes were manually checked, and the automated annotation was corrected if needed. Contigs from the annotated clusters were used to build a repeat library. To minimize potential conflicts due to the occasional presence of contaminating sequences in the clusters, only contigs with average read depths ≥ 5 were included and all regions in these contigs that had read depths < 5 were masked. Genome assemblies were then annotated using custom RepeatMasker search with options '-xsmall -no_is -e ncbi -nolow'. Output from RepeatMasker was parsed using custom scripts (https://github.com/kavonrtep/repeat_annotation_pipeline) to remove overlapping and conflicting annotations.

Tandem repeat annotations were performed using TAREAN tool available from the RepeatExplorer2 output. Consensus monomers were then used as bait to annotate the presence and overall distribution of satellite DNA repeats in the assembled genome using the annotation tool available in Geneious R9[102].

Since we noticed that a few highly repetitive regions were not annotated we additionally used Satellite Repeat Finder (srf,[103], commit: faf9c19) for annotation. We first generated a k-mer distribution of the genome assembly using kmc ([104], v 3.2.1) and then used srf in combination with minimap2 ([96], v2.24) to identify regions containing regions with high k-mer abundance. We then manually inspected all regions where srf resulted in an additional annotation and added them to the RepeatExplorer2 annotation.

### Long read Oxford Nanopore sequencing

Several adult animals representing different sizes and biological conditions (i.e. starved for either 2 weeks or 1 month), regenerating fragments at several stages (from 0 to 7 days after cut) and isolated heads and tails were pooled in order to maximize transcriptomic diversity. Total RNA was extracted from snap-frozen planarian tissue using the protocol described in ref. 105 After the phenol-chloroform extraction step, RNA was purified using a Clean & Concentrator-25 kit (Zymo). Since read size distribution in Nanopore sequencing is usually biased towards the shorter transcripts, we employed the manufacturer's protocol variant optimized for the enrichment of transcripts longer than 200nts. RNA quality and quantity were assessed using Bioanalyzer RNA 6000 Nano Kit (Agilent). The poly-A+ fraction of RNA was isolated using Oligo d(T)$_{25}$ Magnetic Beads (New England BioLabs Inc.) following the commercial protocol for Mammalian Cells provided by the manufacturer. Briefly, 14 μg of total RNA were diluted into 250 μL of Lysis/Binding buffer and used as input of the isolation procedure. 50 μL of oligo-dT beads were employed for each round. After 2 rounds of isolation, the resulting poly-A + RNA fraction (corresponding to 0.7–2% of the starting amount) was then purified again on a Clean & Concentrator-5 Column Kit (Zymo) and eluted in 10 μL of molecular-grade water.

The direct RNA and cDNA libraries for Oxford Nanopore Sequencing were prepared using the SQK-RNA002, SQK-PCS109, and SQK-PCS111 kits, starting from 100 ng and 4 ng of poly-A + RNA, respectively, following the manufacturer's instructions. Sequencing was performed on the Oxford Nanopore Technologies (ONT) platform using a MinION and a PromethION P24 device. The prepared library was loaded onto a R9.4.1 flow cell, and sequencing was initiated following the manufacturer's instructions. Real-time data acquisition was monitored using the ONT sequencing software MinKNOW.

## Genome annotation

The transcript annotation was generated by a hybrid genome-guided approach relying on both 1) dedicated long-read Nanopore cDNA/dRNA sequencing runs and 2) Illumina short-read and poly-adenylation (3P-seq) data obtained by publicly available datasets.

After read quality trimming, deduplication, filtering, and mapping (using HISAT2[106] and minimap2[96] for short and long reads, respectively), a draft transcriptome was generated using Stringtie2[107] then it was further refined using FLAIR[108] and a collection of custom scripts to filter high confidence isoforms. For details of the procedure and a step-by-step guide to the genome annotation analysis see Supporting Information: Section 5. To designate a high-confidence gene set we applied additional filters using the repeat annotation, analysis of transcript expression across all nanopore data using the Nanocount program[109], and a requirement for BLAST homology of at least 75% identity and 75% coverage against either the dd_smed_v6 or dd_smes_v1 transcriptomes, along with considerations for the open reading frame (ORF) length. We excluded all transcripts that overlapped more than 75% with a repeat annotation. For those transcripts with an ORF of 100 amino acids or more, we set a minimum expression threshold of 0.001 Transcripts per Million (TPM). In the case of transcripts with ORFs smaller than 100 but at least 50 amino acids in length, they were included in the high-confidence set under two conditions: if they had a BLAST hit and expression of at least 1 TPM, or in the absence of a BLAST hit, if their expression was at least 10 TPM.

## Benchmarking of *S. mediterranea* annotations

We compared our gene annotations with several transcriptomes that are commonly used in the field. Namely, dd_v1 a non-stranded de novo transcriptome assembly of the sexual strain of *S. mediterranea* (dd_Smes_v1,[42]); dd_v6 a de novo transcriptome assembly of the asexual strain of *S. mediterranea* (dd_Smed_v6,[42]); SMESG an ab initio gene prediction on basis of the previous dd_smes_g4 *S. mediterranea* genome assembly[42]; Oxford_v1 a composite annotation of[38] and[45] in combination with SMESG. We assessed the BUSCO content of the assemblies using BUSCO (v5.0.0), with the 954 genes in the metazoa_odb10 lineage dataset. We ran BUSCO on the transcript level and merged the Complete and Duplicated category. We assessed how published transcripts were represented in the transcriptomes by aligning them to all transcripts available in NCBI using minimap2. To assess the mappability of these transcriptomes, we used sequencing data generated post-July 2020, which was after the creation of the benchmarked annotations. This approach was taken to avoid any bias that might arise from including reads used in the benchmarking process in the data creation. We utilized 13 sequencing libraries that varied in read lengths (152–302 bp), sequencing platforms, and biological conditions (encompassing whole worms, dissociated cells, sorted X1 cells, and regenerating wound regions) to provide a comprehensive representation of common conditions in the field. The mapping efficiency was assessed using the BWA tool. Finally, we mapped all transcripts to the schMedS3h1 assembly using minimap2 and manually inspected 96 gene models for frame shifts, truncations, chimeras, and fragmentation.

## Nuclei isolation for ChIP-seq and Hi-C

For *S. mediterranea* 100 worms of ~7 mm were treated with NAC for 10 minutes and afterwards rinsed with dH2O. The animals were transferred into a 15 mL Dounce tissue grinder and all excess water was removed. 10 mL modified cell buffer (20 mM HEPES Hemisalt, 25 mM NaCl, 0.85 mM KCl, 1.5 mM EDTA, 1.5 mM EGTA) (+1x Halt™ProteaseInhibitor + 10 mM NaButyrate) containing 1% methanol-free formaldehyde (FA) was added and a timer set to 10 minutes. The tissue was homogenized using pestle A (clearance 0.0035–0.0065 in.) till resistance was minimal and incubated on a rocking shaker. Cross-linking was stopped by adding glycine (125 mM glycine per 1% FA in

10 mL fixation buffer) and incubated for 5 minutes at room temperature. The sample was centrifuged in a swing bucket centrifuge for 10 minutes at 1000 x g at 4 °C. From this point on all steps were conducted at 4 °C unless stated otherwise. The supernatant was gently removed and the pellet was resuspended in 10 mL of Buffer A (20 mM HEPES hemisodium salt, 25 mM NaCl, 10 mM EDTA, 0.5 mM EGTA, 0.25% TritonX-100, 0.5% Igepal CA-630) (+1x Halt™ ProteaseInhibitor + 10 mM NaButyrate). Nuclei release was supported by mechanical disruption using pestle B (0.0010-0.0030 in.) till resistance was minimal. The sample was transferred into a 15 mL tube and incubated for 15 minutes on ice on a rocking shaker. After pelleting at 1000 x g for 10 minutes at 4 °C, the supernatant was gently removed and the pellet was resuspended in 10 mL of Buffer B (20 mM HEPES hemisodium salt, 0.2 M NaCl, 1 mM EDTA, 0.5 mM EGTA) (+1x Halt™ ProteaseInhibitor + 10 mM NaButyrate) and incubated shaking vertically for 15 minutes on ice. The sample was then filtered through a 50 µm mesh and an aliquot of 100 µL was used for nuclei counting. The remaining solution was centrifuged at 1000 x g for 15 minutes at 4 °C, then the supernatant was removed. The nuclei pellet was resuspended in 1 mL Buffer B and the nuclei were distributed into $2 \times 10^7$ aliquots for ChIP-seq experiments or $1.1 \times 10^6$ nuclei aliquots for Hi-C. Finally, the samples were centrifuged at 1050 x g for 10 minutes at 4 °C, supernatant was removed and the pellets snap frozen and stored at −80 °C.

## SDS-PAGE and Western Blot

Pulldown samples were directly mixed with 6x Laemmli buffer (12% (w/v) SDS, 0.06% (w/v) Bromophenol blue, 50% Glycerol (w/v), 600 mM DTT, 60 mM Tris-HCl pH 6.8), decrosslinked and denatured for 15 minutes at 95 °C. The unbound fraction was concentrated via acetone precipitation. Four volumes of cold (−20 °C) acetone were added to the sample, mixed, and incubated for 60 minutes at −20 °C. Then the sample was centrifuged for 10 minutes at 15,000 × g and the supernatant was removed. The protein pellet was finally resuspended in the same volume as the pulldown sample, mixed with 6x Laemmli buffer and decrosslinked and denatured at 95 °C for 15 minutes. Samples were run on NuPAGE Novex 4–12% Bis-Tris protein gels in 1x MOPS running buffer, transferred onto Nitrocellulose membranes in transfer buffer (1x MOPS with 20% (v/v) MeOH). The membrane was blocked in 1x PBS with 0.1% (v/v) Tween20 and 5% (w/v) nonfat dry milk and incubated with the primary antibody (α-H3K4me3 - merckmillipore Cat.#07-473 Lot#3381394) diluted 1:5000 in 1x PBS with 0.1% (v/v) Tween20 and 5% (w/v) nonfat dry milk. Membrane was washed with washing buffer (1x PBS with 0.1% (v/v) Tween20) prior to incubation with fluorescent secondary antibodies (anti-Rabbit IRDye 800CW, LICOR Cat.#926-32213) diluted 1:20,000 in blocking solution. The membrane was washed with washing buffer, followed by a final washing step in 1x PBS without Tween20. The stained membrane was dried and imaged on a LI-COR Odyssey imager.

## Chromatin immunoprecipitation and sequencing (ChIP-seq)

The frozen nuclei pellet was resuspended in 0.333 mL of lysis buffer (50 mM Tris-HCl pH7.5, 10 mM EDTA, 1% (w/v) SDS) (+1x Halt™ ProteaseInhibitor + 0.5 mM NaButyrate) and incubated for 25 minutes at 4 °C while rocking. Then the sample was transferred into a milliTUBE with AFA fiber (Covaris Part# 520135) and topped up with Dilution buffer A (50 mM Tris pH7.5, 10 mM EDTA) (+1x Halt™ ProteaseInhibitor + 0.5 mM NaButyrate) to dilute SDS to ~0.3%. The sample was sonicated using the Covaris S220 Focused-ultrasonicator at 140 Peak Power, 5.0 Duty Factor and 200 Cycles/Burst for 15 minutes at maximum 8 °C, then transferred into a fresh 1.5 mL tube and centrifuged for 5 minutes at 4 °C (10,000 x *g*) to pelletize debris. The supernatant was split into aliquots of 150 µL and subsequently used for pull-down experiments. For input, one aliquot was topped up with Buffer B (20 mM HEPES hemisodium salt, 0.2 M NaCl, 1 mM EDTA, 0.5 mM EGTA) (+1x Halt™ ProteaseInhibitor + 10 mM NaButyrate) to 475 µL

and 20 μL 5 M NaCl and 5 μL proteinase K (20 mg/mL) were added and incubated at 65 °C overnight to reverse cross-links. The next day the sample was removed from the thermoblock and cooled down to room temperature. Subsequently, 2 µl RNase (10 mg/mL) was added and incubated for 60 miutes at 37 °C. DNA was isolated using the PCI-DNA extraction method. The sample was transferred into a phase lock tube, 500 µL PCI-mixture was added to the sample, vortexed, and centrifuged at full speed for 5 minutes at room temperature. 500 µL of pure chloroform was added, vortexed, and centrifuge at full speed for another 5 minutes at room temperature. The upper aqueous phase was transferred into a 1.5 mL tube, 500 µL isopropanol (−20 °C), 100 µL 3 M sodium acetate and 2 µL glycogen were added and then incubated for 60 minutes at −20 °C. After incubation the sample was centrifuged at full speed for 15 minutes at 4 °C and then the supernatant was removed. The pellet was washed with 500 µL 96% ethanol (−20 °C) and centrifuged at full speed for 10 minutes at 4 °C. A second washing step was performed with 500 µL 70% ethanol (−20 °C) and then centrifuged at full speed for 10 minutes at 4 °C. After removing the supernatant, the pellet was dried at 55 °C. The DNA was resuspended into 50 µL EB (50 mM Tris-HCl pH8) buffer and stored at −20 °C before library preparation.

For pull-down one aliquot was used per target. 2 Volumes of Dilution buffer B (50 mM Tris-HCl pH7,5, 225 mM NaCl, 0.75% Sodium Deoxycholate, 1.5% NP-40) were added to the sample and topped up to 1 mL with RIPA buffer (50 mM Tris-HCl pH7,5, 150 mM NaCl, 0.1% SDS, 0.5% Sodium Deoxycholate, 1% NP-40) ( + 1x Halt™ ProteaseInhibitor + 0.5 mM NaButyrate). 2.5 µL α-H3K4me3 (merckmillipore Cat.#07-473 Lot#3381394) or 5 µL α-H3K27ac (activemotif Cat.#39133 Lot#16119013) from the stock solution was added and incubated for 60 minutes at 4 °C with gentle agitation (6 rpm). Subsequently, 30 µL Magna ChIP Protein A Magnetic Beads were added to the sample and incubated at 4 °C with gentle agitation (6 rpm) overnight. Beads were accumulated using a magnetic rack. Washing was performed by adding 1 mL washing solution, incubating with gentle agitation for 5 minutes at 4 °C, and removing solution. The following buffers were used for washing; RIPA (50 mM Tris-HCl pH7,5, 150 mM NaCl, 0.1% SDS, 0.5% Sodium Deoxycholate, 1% NP-40), HiSalt (50 mM Tris-HCl pH7,5, 500 mM NaCl, 0.1% SDS, 1% NP-40), LiCl (50 mM Tris-HCl pH7,5, 250 mM LiCl, 0.5% Sodium Deoxycholate, 1% NP-40), TE (10 mM Tris-HCl pH7,5, 1 mM EDTA) (2 times). After the final wash, the TE buffer was thoroughly removed and the bead-bound complexes released by incubating in 100 µL elution buffer (50 mM Tris pH7,5, 10 mM EDTA, 1% SDS, 10 mM DTT) at 4 °C for 30 minutes. The supernatant was transferred into a fresh tube and continued with reversing cross-links and DNA isolation, as stated above.

Immuno-precipitated DNA samples at an input amount of 2–100 ng were subjected to Illumina fragment library preparation using the NEBnext Ultra II DNA library preparation chemistry (New England Biolabs, E7370L). In brief, DNA fragments were end-repaired, A-tailed, and ligated to unique-dual indexed Illumina TruSeq adapters. Resulting libraries were PCR-amplified for 15 cycles using universal primers (Primer 1: CAAGCAGAAGACGGCATACGAGAT and Primer 2: AATGA-TACGGCGACCACCGA*G; *: Phosphothioate bond), purified using XP beads (Beckman Coulter) with a bead to library ratio of 1:1. The libraries were size selected using XP beads with a 0.6:1 right and 1:1 left bead to library ratio and if needed subjected to an extra 0.8:1 bead to library ratio to remove the left over adaptor dimers. They were checked for their quality and quantified using Fragment Analyzer (Agilent). Final libraries were subjected to 100-bp-paired-end sequencing on the Illumina NovaSeq6000 and 75-bp-paired-end and single-end on the NextSeq500 platform to a depth of 30-70 million fragments per library.

## Chromatin conformation capture (Hi-C)

Chromatin conformation capturing was done using the ARIMA-HiC High Coverage Kit (Article Nr. A101030-ARI) following the Arima

documents (User Guide for Animal Tissues, Part Number A160162 v00). $1 × 10^6$ crosslinked nuclei from each species went into lysis step. The crosslinked gDNA was digested with a cocktail of four restriction enzymes. The 5′-overhangs were filled in and labelled with biotin. Spatially proximal digested DNA ends were ligated, and the ligated biotin-containing fragments were enriched and went for Illumina library preparation, following the ARIMA user guide for Library preparation using the Kapa Hyper Prep kit (ARIMA Document Part Number A160139 v00). The barcoded Hi-C libraries ran on an S4 flow cell of an Illumina NovaSeq 6000 with 2 × 150 cycles.

## Assay for transposase-accessible chromatin with sequencing (ATAC-seq)

The ATAC-seq experimental protocol is a modification of the protocol from[46]. For each biological replicate of the wt and x-ray condition 10 worms of ~7 mm were treated with NAC for 10 minutes and afterwards rinsed with dH2O. The animals were transferred into Covaris tissue-TUBEs (TT05M TX), snap frozen in liquid nitrogen and crushed with setting 2 using the Covaris CryoPrep (CP02) device. Snap freezing and crushing was repeated a second time, before the powder was stored at −80 °C. For nuclei isolation two corresponding samples were processed simultaneously (wt and x-ray). The sample was resuspended in 1.5 mL of cold 1x homogenization buffer (20 mM Tris-HCl pH8, 0.25 M sucrose, 30 mM KCl, 10 mM $MgCl_2$, 0.3% (v/v) Igepal CA-630, 1 mM DTT, 0.5 mM Spermidine, 0.25 mM Spermidine, 1x Halt™ ProteaseInhibitor) and transferred into a pre-chilled tissue grinder. The powder was further homogenized with a Dounce homogenizer and pestle B (clearance 0.0005-0.0025 in.) on ice with 10 strokes and afterwards filtered through a 50 µm mesh. 1.5 mL of 50% iodixanol solution (20 mM Tris-HCl pH8, 50% (v/v) Iodixanol, 25 mM KCl, 5 mM $MgCl_2$, 0.5 mM Spermidine, 0.25 mM Spermidine, 1x Halt™ ProteaseInhibitor) was added and well mixed.

For gradient centrifugation 2000 µL of a 40% iodixanol solution (20 mM Tris-HCl pH8, 40% (v/v) Iodixanol, 26 mM KCl, 6 mM $MgCl_2$, 0.5 mM Spermidine, 0.25 mM Spermidine, 1x Halt™ ProteaseInhibitor) was transferred into a 15 mL tube. 1000 µL of 30% iodixanol solution (20 mM Tris-HCl pH8, 30% (v/v) Iodixanol, 27 mM KCl, 7 mM $MgCl_2$, 0.5 mM Spermidine, 0.25 mM Spermidine, 1x Halt™ ProteaseInhibitor) was slowly layered on top of the 40% mixture. Finally, the 25% iodixanol-nuclei solution was carefully added on top of the 30% mixture. Separation was performed by centrifugation at 3000 x $g$ with brakes off at 4 °C for 20 minutes. Then, the nuclei band was collected and transferred to a fresh tube. 2 volumes of ATAC-RSB-Buffer (20 mM Tris-HCl pH8, 10 mM NaCl, 3 mM $MgCl_2$ + 1 mM DTT) were added before nuclei were counted. For each biological replicate 3 libraries were prepared for tagmentation. Therefore, $5 × 10^4$ nuclei were transferred into separate tubes and centrifuged at 900 x g for 7 minutes at 4 °C. The supernatant was discarded, and the pellet was resuspended in 25 µL ATAC-seq reaction mix (0.1% Tween-20, 0.01% Digitonin, 1x TD Buffer (2x TD Buffer: 20 mM Tris base, 10 mM MagCl2, pH 7.6 with acetic acid; 20% freshly added Dimethylformamide) + 1 mM DTT), then 25 µL commercial buffer from "Illumina Tagment DNA TDE1 enzyme and buffer kit" with Tn5 Transposase enzyme was added and the mixture was incubated for 30 minutes at 37 °C shaking at 1000 rpm. Tagmentation was stopped by proceeding with the Zymo PCR Clean and Concentrate kit. Finally, tagmented DNA was eluted in 10 µL and used for library amplification.

10 µL of purified tagmented DNA was indexed and pre-amplified for initial 5 PCR cycles with 1x KAPA HiFi HotStart Readymix and 100 nM unique dual index P5 and P7 primers compatible with Illumina Nextera DNA barcoding, under the following PCR conditions: 72 °C for 5 minutes, 98 °C for 30 s, thermocycling for 5 cycles at 98 °C for 10 s, 63 °C for 30 s and 72 °C for 1 minute. Subsequently, a qPCR on the LightCycler 480 (Roche) was performed with 1 µL of the pre-amplified material to determine the remaining PCR cycle numbers (7–13) to

avoid saturation and potential biases in library amplification (see ref. 110). Purification and double-sided size selection of amplified libraries was done with AMPure XP beads (Beckmann Coulter; starting with a 1.55x volume of XP bead purification, followed by a 0.6x/1.55x double-sided size selection, an additional 0.6x/1.55x double sided size selection was performed if needed), and checked for their quality and quantity on a Fragment Analyzer (Agilent). Libraries were sequenced on the Illumina NovaSeq 6000 with PE 100 bp reads to a depth of 40-150 M read pairs.

### ATAC-seq and ChIP-seq read processing and mapping

ATAC-seq and ChIP-seq reads were trimmed using trim_galore (v0.6.6) in --paired mode and QC was done using fastqc (v0.11.9) prior to mapping. Genomes were indexed using bwa index. Due to the size limitations for BAI indexing we split Chromosome 1 at position 414,900,000 and 166,500,000 of *S. lugubris* and *S. nova*, respectively. Of note, in the proximity of these positions no gene annotations were observed. Trimmed reads were mapped to the corresponding genome using bwa (v0.0.17) with the following command "bwa mem -M". Mapped reads were filtered using samtools fixmate -m and samtools sort, before PCR duplicates were removed using Picard (v2.25.5) with the following setting MarkDuplicates REMOVE_DUPLICATES=True VALIDATION_STRINGENCY = LENIENT. Finally reads were filtered using samtools view samtools view -h -b -q 20 -f 3. Libraries sequenced on the different sequencing chips, were merged after this step using samtools merge and subsequently sorted an index as previously described.

### ChIP-seq analysis

Peak calling for ChIP-seq data was performed using MACS2 (v2.2.7.1[111]) running macs2 callpeak -t *SAMPLE* -c *INPUT* -f BAMPE --nomodel --bdg --keep-dup all -g 6.44e8. Effective genome size was calculated using the number of uniquely mappable bases using a k-mer based approach with the khmer software[112] and a k-mer size of 32 and a read length of 100 bp. Peak annotations were generated with the ChIPseeker library (v1.21.1 and v1.33.2[113]) utilizing the annotatePeak function, defining Promoter as 3000 – 0 upstream of the TSS. Profile plots and heatmaps were generated using deeptools (3.5.1 and 3.5.2,[114]) package using different tools and modified manually for better readability. Signal tracks were generated using SparK.py (v2.6.2[115]). Intersections of different regions were done using bedtools (2.30.4,[116]). To quality control the ChIP-seq data we assigned the associated genes into quartiles of gene expression in a dataset of 7 mm-sized wild-type asexual *S. mediterranea*. To quantify expression we used RNA-seq reads, quality trimmed using trimmomatic (v0.32[117]), mapped and quantified using STAR (v2.7.10[118]) and normalized to transcripts per million per sample. We then used the mean of both samples as the expression estimate of that gene.

### ATAC-seq analysis

ATAC-seq QC was performed using the ATAC-seq QC library (1.16.0 and 1.22.0,[119]). Fragment size distribution was calculated and visualized with the fragSizeDist function. For the distribution of fragments corresponding to nucleosomal-free and mononulceosomal regions, the enriched fragments function was used. Technical replicates were merged using samtools merge prior to further processing. Peak calling was performed with MACS2 using the following parameters: -f BAMPE --keep-dup all and species-specific effective genome size with the -g parameter. Peak sets of the same biological condition were combined using ChIP-R (1.23.5[120]) with the parameter --minentries 3. Summit information was added to the generated files by averaging the positions of the 3 original summits using a custom script. Final consensus peak sets for each species were generated using bed tools intersect by adding condition-specific peaks to the wild-type peak set. Signal tracks were generated using SparK.py.

### Phylogenetics

To generate a phylogenetic hypothesis, we used Orthofinder (v2.5.5[121]) to determine single-copy orthologs between the *Schmidtea* and parasite transcriptomes. Then we aligned each orthogroup using MAFFT (v7.525[122], command: '--ep 0 --genafpair –max iterate 1000') and determined the best fitting amino acid substitution model using ModelFinder[123] with the BIC criterion. Finally, we inferred a consensus maximum likelihood phylogeny with IQ-TREE (v2.3.0[124]) using the concatenated amino acid alignment and the best-fitting substitution model for each partition. We assessed branch support using 1000 ultra-fast bootstraps and 1000 replicates of the SH-like approximate likelihood ratio test.

### Sequence divergence

To determine neutral evolutionary distance across *Schmidtea*, we generated a reference-free whole-genome alignment using Progressive Cactus (v2.6.9[125]). Initially, we soft-masked each genome utilizing our custom repeat annotations. Following this, we conducted the whole-genome alignment with the default settings. The output HAL file was then converted to the MAF format using the cactus-hal2maf tool. We then generated evolutionary distance matrices for 4-fold degenerate sites, assuming the phylogenetic tree previously inferred (see above) using the PhyloFit program from the PHAST package version included with Progressive Cactus.

### Evolutionary conservation of ATAC-seq peaks

We used our whole-genome alignment to determine if the sequences contained in an ATAC-seq peak and their chromatin state was conserved across *Schmidtea*. Existing methodologies for regulatory region conservation assessment using Progressive Cactus alignments (e.g. ref. 126), were observed to inadequately deal with the high extent of fragmentation and duplication in planarian genomes. Additionally, they do not leverage ATAC-seq data in the recipient species to filter duplicated peak regions. Hence, a custom protocol was developed using a series of scripts to address the inadequacies associated with peak fragmentation and duplication. The base dataset was constituted by the set of consensus ATAC-seq peaks (see ATAC-seq data analysis). The *S. mediterranea* pseudohaplotype 1 served as the frame of reference in the subsequent steps. The peak region (RGN) and a 51 bp extension of the summit (SUM) for each species were processed using halLiftover[127]. This allowed for an independent assessment of the presence of the summit in the recipient species. Any RGN liftover regions less than 20 bp in size were disregarded, and fragmented RGN liftover regions separated by less than 100 bp were merged. RGN liftover regions that overlapped with the corresponding SUM liftover were identified as 'conserved regions'. Those RGN liftover regions without a SUM overlap were classified as 'not conserved'. The conserved regions were characterized by a median size of 555 bp, in contrast to the short, highly fragmented size of the non-conserved liftovers (Supporting Information: Section 3), indicating the efficacy of the developed pipeline in filtering out low-confidence liftovers. Conserved regions were then examined for an overlap with the ATAC-seq signal in the recipient species. A liberal scoring criterion was employed, treating any overlap as a hit and marking conserved regions with an overlap as a 'conserved peak'. Finally, the results of the conservation assessment in each species were used to annotate each *S. mediterranea* pseudohaplotype 1 peak. The highest conservation status was given to peaks that showed conservation in all three recipient species.

### Synteny analysis

Since we expected gene order to be largely preserved across *Schmidtea*, we used the R package GENESPACE (v1.0.8,[63]) to infer gene-order-based syntenic blocks. GENESPACE uses a combination of Orthofinder and a reimplementation of the MCScanX algorithm[128]. We then

visualized the resulting syntenic blocks using the built-in riparian plot function. To understand the conservation of synteny across increasingly larger phylogenetic distances, we employed three tools and four approaches. First, we annotated BUSCO genes present in *Schmidtea* and the parasites using BUSCO (v5.0.0,[129])and then identified those genes that were present in each species as a single copy gene. We then visualized the location of the BUSCOs in each assembly by coloring them according to the chromosomal location in *Schistosoma mansoni*. Second, we used Orthofinder (v2.5.5,[121]) with default parameters and with protein sequences representing the longest isoform of the entire protein-coding fraction of the *Schmidtea* and the parasites annotations as input. We then processed the resulting orthogroups for each pairwise combination always only retaining pairwise single copy orthologs. Thus, for each pair the number of orthologs used differed. We visualized the distribution of orthologs using a dotplot and conducted chi-squared tests against the null hypothesis that orthologs are randomly distributed across the chromosomes. A significant p-value indicates that there is a clustering of orthologs based on the chromosome combination (i.e. preservation of synteny). Given potential violations of chi-square test assumptions by our genomic data, we not only conducted a conventional significance test based on the chi-square distribution but also employed a permutation test with 100,000 permutations to assess significance. We calculated the effect size 'Cramer's V' to determine the amount of clustering. Cramer's V is 0 for a random distribution and 1 for a perfect correlation between chromosomes. Third, we used the reciprocal-best-blast and Fisher's exact tests approach as implemented in the ODP tool[10] (v0.3.0) to infer pairwise orthologs between species and test for synteny conservation. Given these analyses test for a direct association between two chromosomes/scaffolds, we included the highly contiguous, albeit not chromosome-scale, genome assembly of *Macrostomum hystrix*. Finally, we assessed if ancestral metazoan linkage groups (MALG), i.e. linkage groups that are present in bilaterians, cnidarians, and sponges, defined in ref. 8 were preserved in the flatworm genomes using the ODP tool. The tool uses hidden-markov models to identify homologs to the MALG proteins and tests for their enrichment on particular chromosomes using Fisher's exact test. We then summarized the results using the chromosome tectonics algebra defined in ref. 8.

### Synteny breakpoint enrichment

We investigated whether synteny breakpoints were disproportionately associated with specific repetitive sequences using the R package GenomicRanges (v.1.46.1). For each syntenic block, delineated by GENESPACE, across *Schmidtea* species, we established 10 kb flanking windows in both genomes involved in the pairwise combinations. Subsequently, we evaluated their enrichment by contrasting them against 1000 iterations of random placements of an equivalent number of windows. We performed a two-tailed test to assess if the observed elements were present in higher or lower quantities than expected from the random iterations. To account for multiple testing, we adjusted the p-value for all tested elements with at least 10 members on average (this threshold was used to prevent loss of statistical power when testing elements that were exceedingly rare) utilizing the False Discovery Rate method.

### Reporting summary

Further information on research design is available in the Nature Portfolio Reporting Summary linked to this article.

## Data availability

The whole-genome, Hi-C, ATAC-seq, ChIP-seq, and RNA-Seq of *Schmidtea mediterranea*, *Schmidtea polychroa*, *Schmidtea nova*, and *Schmidtea lugubris* data generated in this study have been deposited in the NCBI database under accession code PRJNA1052007. The repetitive element annotation of *Schmidtea* genomes data generated in this study have been deposited in the Zenodo database under accession code 11004547. The *Clonorchis sinensis* genome and annotation data used in this study are available in the NCBI database under accession code PRJNA386618. The *Schistosoma mansoni* genome and annotation data used in this study are available in the NCBI database under accession code PRJEA36577. The *Taenia multiceps* genome and annotation data used in this study are available in the NCBI database under accession code PRJNA307624. The *Hymenolepis microstoma* genome and annotation data used in this study are available in the NCBI database under accession code PRJEB124. The *Macrostomum hystrix* gene annotation data used in this study are available in the Zenodo database under accession code 7861770. The *Macrostomum hystrix* genome data used in this study are available in the European Nucleotide Archive database under accession code GCA_950097015. Source data are provided with this paper.

## Code availability

The code used to conduct the analysis in this study is available at https://github.com/Jeremias-Brand/PlanarianGenomeAnalysis and has been archived in the Zenodo database under accession code 13123038.

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

## Acknowledgements

We thank Heino Andreas, Rick Kluivert, Jens Krull, and the MPI-NAT animal services staff for worm care and the maintenance of the species collection. We thank Tobias Boothe for picture acquisition. We thank Juliana G. Roscito for sequencing support and close collaboration during method development. We thank Hanh Vu for access to *S. medi-terranea* RNA-seq data. We thank the following facilities for their support: the DRESDEN Concept Genome Center, part of the MPI-CBG and the technology platform of the CMCB at the TU Dresden, supported by DFG (INST 269/768-1); the Genomics Core Facility of the European Molecular Biology Laboratory in Heidelberg; and the IIT Genomics Facility. We are grateful to Diego Vozzi, Yeraldin Castillo Spelorzi, and Edoardo Henzen for their technical support of the Nanopore sequencing experiments. JCR received funding from the European Research Council (ERC) under the European Union's Horizon 2020 research and innovation program (grant agreement number 649024), from the German Research Foun-dation (project RI 2449/51), from the Behrens-Weise Foundation, and from the Max Planck Society. JNB was supported by Swiss National Science Foundation Grant P500PB_206673. LP, AC and SG were sup-ported by intramural funding of the Istituto Italiano di Tecnologia. AP was supported by the SPP2202 Priority program (Project No. 422389065).

## Author contributions

M.I., J.N.B, and J.C.R designed the study and wrote the manuscript. T.B., M.P., M.Z., and A.M. performed genome assembly. L.P., A.R., L. R., and J.N.B performed genome annotation. T.S. and M.A.G. developed and applied DNA extraction protocols. M.I. designed, developed, performed, and analyzed ATAC-seq and ChIP-seq experiments. S.W. performed method development and application of HiFi sequencing. M.I. per-formed Hi-C experiments. S.Z. and A.P. supported and performed some of the Hi-C experiments. L.P., A.C., and S.G. performed nanopore sequencing. M.V.F. and J.C.R. collected and identified *S. nova*, *S. poly-chroa*, and *S. lugubris*. J.N.B. performed genome alignment, phyloge-netics, and comparative genomics analyses. J.C.R. received funding and supervised the research.

## Funding

## Competing interests

The authors declare no competing interests.
