## [Peer Review File · Nature Communications]

REVIEWER COMMENTS

Reviewer #1 (Remarks to the Author):

First, I would suggest acceptance with minor revisions. This paper is quite good and is not missing much. Only a few comments need to be addressed as listed below.

Comments:

Line 270: 20% of enhancers designated as “distal intergenic” is used as evidence that *S. mediterranea* regulatory elements are close to genes, but there is no evidence presented that 20% is unusual. The paper would benefit from a distal vs proximal comparison to other organisms such that we know if this is highly unusual or not.

Line 276: Given that the whole-animal averaged peak annotations are likely a subset of “constitutively expressed genes or those active in the most abundant cell types. “, is this a sufficient set to support the claim that *S. mediterranea* regulatory elements tend not to be distal? Do constitutively active and highly expressed genes in other species normally have distal enhancers, or are these genes that tend to have local regulatory regions?

Genome Architecture and Synteny: BUSCO is used for gene synteny across species including parasitic flatworms and it is suggested that a clade specific BUSCO data set would be useful as there are 125 missing BUSCOs in all 4 flatworms studied. Given that BUSCO is used for the synteny analysis with parasitic flatworms, it would be nice to know if these missing BUSCOs are also missing in parasitic flatworms.

Citations: Several bioinformatics tools are missing citations, including trimmomatic, STAR, samtools, SALSA, PICARD, bwa-mem, pbmm2 and DeepVariant. There may be others that were missed. Please check and make sure all tools/contributions are appropriately cited.

Methods: No mention is made of which BUSCO lineage was used for the figures or analyses; this should be explicitly stated.

Reviewer #2 (Remarks to the Author):

Ivankovic et al present the genome assembly and annotation of the flatworm *Schmidtea mediterranea*, and three other sister species in *Schmidtea*, as well as an analysis of gene regulatory regions based on ATAC-seq and ChIP-seq data. They find that promoter- and enhancer-associated histone marks and open chromatin regions are broadly conserved across the genus, and that gene order (synteny) relative to other animals has been largely lost in *Schmidtea*. These findings, in addition to opening new research directions into how flatworm genomes are evolving, will provide a valuable genomics resource to the research community. This is a strong contribution and very interesting manuscript. There are no major flaws; the points below can be considered for further strengthening/clarifying.

In Fig 1, the authors benchmark the new 'S3BH_all' gene annotations (and others such as the commonly used dd_v6 transcriptome assembly) against various standards such as published transcripts from genbank, BUSCO, etc. It would also be helpful to see a direct comparison to dd_v6, given that so many planarian labs use this particular version. Are the S3BH_all transcripts a superset of the dd_v6 transcripts, or are there some genes that are missing? In cases when they disagree, does it appear there is something wrong with the dd_v6 transcript? An actual list of the disagreements, grouped by type, and some sort of figure panel summarizing the list might be helpful.

One of the fascinating results of this paper is that synteny with the Metazoan Ancestral Linkage Groups (MALGs) has been lost independently in more than one platyhelminth lineage. Based on this the authors suggest that gene positioning might not be important in flatworms. The data seem consistent with this idea. However, I can also imagine a situation where genes cluster on the chromosomes according to their function, and where clustering itself is conserved despite exchangeability of gene position within and between clusters. The authors could examine whether genes cluster spatially in the various species according to their function, for example by looking at Gene Ontology (GO) terms. Alternatively, the authors could limit 'gene positioning may not be essential' to a claim that conservation of synteny is not essential (which the data more strongly indicate).

Methods Section: I can't find a description of how the whole-genome alignments were done. Did the authors use Progressive Cactus or something like that?

Data Access Section: Please remember to make the repetitive element annotations publicly available in gff format or equivalent.

Figure 1A: The lack of long-range contacts in the Hi-C data is curious, so readers might benefit from some more commentary about this in the discussion section. Guo et al (2022) saw something similar in their Schmidtea Hi-C data, so evidently it is reproducible, but I don't see much commentary except on lines 542-543. How common is this phenomenon in other species? Please add a reference or two.

Figure 1B: Mercury spectra-cn plots should be added to the supplement. Some things like spurious duplication levels are not (as I understand it) captured in the quality value or completeness metrics alone and it can be diagnostic to look directly at the spectra-cn. Not necessarily for all four read data sets, but perhaps a representative one or two.

Figure 1H: On the x-axis the label "dd_v1.PCF" is repeated and "dd_v6.PCF" is missing, so one of the bars must be mis-labeled.

Figure 2E: A Venn Diagram seems more appropriate than a bar chart, especially for the rightmost bar, which requires an awkward sort of legend to explain (which is too small to be read easily).

Lines 270: "indicates" could be changed to "suggests". Given the data, it's a reasonable hypothesis that regulatory elements are close to genes in Schmidtea, but more data are probably needed to use "indicates".

Figure 2G: Do these proportions look different if the enhancers/promoters are stratified according to whether or not their associated gene is variably expressed across tissues in the planarian single-cell sequencing atlas? Housekeeping genes usually comprise a sizeable chunk of the transcriptome, and I'd suppose they'd also show strong ChIP-seq peaks, so any tissue-specific signal might be obscured. If you need more room to extend 2G to accommodate this, I'd remove 2I, which seems redundant.

Figure 2H-I and Figure 3H: The x-axis should be labeled in kilobases relative to the TSS in these plots.

Figure 3H: The TSS of the *S. mediterranea* ATAC-seq plot should be aligned with the TSS of the three sister species in the plots below.

Line 288: 'data not shown': Perhaps there is a reference that can be added or some other way of saying this.

Figure 4B: The chromosome order and orientations in the riparian plot should be chosen to minimize the crossing of 'braids' between all species. This will make it easier to understand the evolutionary history. I believe it can be done by increasing the 'syntenyWeight' parameter in the 'plot_riparian()' function of GENESPACE. For instance, crossing is decreased by making the *S. lugubris* chromosome order 1-3-2-4 instead of 1-2-3-4.

Figure 4H-J: Based on Figure 4D-G, it seems clear that there was independent loss of synteny relative to the Metazoan Ancestral Linkage Groups (MALGs) in both Schmidtea and Macrostomorpha, so in Figure 4H-J the chosen outgroup (*M. hystrix*) is derived, which is not ideal in terms of clarity and ease of interpretation. Could either choose an outgroup for panels H-J that is ancestral for synteny with other animals (perhaps some mollusk?) or move H-J to the supplement (it seems redundant with panels E-G anyway, so maybe the latter is the way to go).

Typographical points:

Lines 211-213: The sentence beginning with "Collectively..." has confusing syntax, and maybe also a typo ("93 out of 91"?). Perhaps it should read: "Collectively, the S3 annotations had the highest fraction of error-free gene models, contained models of all the test genes, and had intact ORF representations for 93 out of 96 test genes."

Line 271: Gene name "sp5" should be italicized.

Line 286: "Namely, all known members of the genus, " Replace with a colon (sentence fragment).

Line 316: "robust" should be "robustness".

Line 450: "significantly improves over" should be "is a significant improvement over" (improve should not be used intransitively here).

Line 550: “artefect” should be “artifact”.

Reviewer #3 (Remarks to the Author):

In this manuscript, Ivankovic and Brand et al. present a haplotype-phased genome assembly for a sexually reproducing strain of *Schmidtea mediterranea*, alongside chromosome-scale genome sequences for three related *Schmidtea* species. Over recent decades, *S. mediterranea* has become a famous model system for stem cell biology. However, it has been challenging to sequence its genome due to its unique biology, such as high AT and repeat content. As a result, the first high-quality genome for a sexually reproducing *S. mediterranea* strain was published in 2018, utilizing long-read sequencing. Subsequently, in 2022, another sexual strain was sequenced to chromosome-scale by the other group using advanced Hi-C scaffolding. This study showed that *S. mediterranea* possesses four chromosomes, with chromosome 1 suggested as a potential sex-primed autosome due to its reduced recombination rate.

This study builds upon previous work by resequencing the *S. mediterranea* genome and sequencing three additional species within the *Schmidtea* genus. In addition, the authors employ ATAC-seq and ChIP-seq to explore the evolution of epigenetic regulation in planarians. The authors discover that the reduced recombination rate in chromosome 1 of *S. mediterranea* is attributable to several large inversions that distinguish between haplotypes 1 and 2. They further conducted comparative analyses with other flatworms to explore the conservation of genome regulation and structure.

The manuscript is commended for its clarity and the breadth of its findings, appealing to a broad audience. I particularly appreciate the effort to contextualize the study within the wider classification of planarians as part of the Platyhelminthes phylum. Despite the widespread recognition of *S. mediterranea* as a model, it is essential to remember that it belongs to a rapidly evolving, derived lineage within the flatworms, marked by distinct biological characteristics and lineage-specific traits. Nonetheless, the current manuscript could be improved by enhancing the comparative genomics and synteny analyses.

Major comments:

1. The manuscript is a pioneering work in comparative genomics within the *Schmidtea* genus. Yet, it appears to overlook the inclusion of a closely related outgroup and the reconstruction of ancestral features, essential elements of comparative genomics. Incorporating a closely related outgroup would significantly enhance the comparative analysis, offering more profound insights into the genomics of *Schmidtea*. In instances where chromosome-scale genomic data may not clarify phylogenetic relationships, considering draft genomes and transcriptome data previously published by the authors could provide valuable additional context.

2. The manuscript would benefit from a clearer explanation regarding which Schmidtea genomes reflect the ancestral state in the synteny analysis, particularly concerning the evolution of the sex-primed chromosome in *S. mediterranea*. Clarifying whether this chromosome resulted from the fusion of chromosomes from different Schmidtea lineages would significantly enhance the narrative and findings. This leads to the following comment.

3. In the section starting around Line 384, the writing leaves some ambiguity regarding the occurrence of 'split' or 'fusion' events in the chromosomes being discussed. As previously mentioned, this uncertainty likely arises from a notable absence of detailed ancestral state reconstruction. For example, a cursory review of Fig. 4B reveals at least seven linkage groups in the ancestral state across all Schmidtea genomes examined. In fact, the detection of gene intermingling between schMedS3h2 chromosome 1 and schPol2 chromosomes 3 and 4 suggests a fusion-with-mixing event may have occurred from schPol2 to schMedS3h2, highlighting a schMed-specific fusion event. This observation supports the notion that fusion-with-mixing events, often considered irreversible and directional, play a crucial role in chromosomal evolution (e.g., lineage-specific fusions on chromosomes 1, 2, and 3 unique to schMed). Elucidating the impact of these events on the development of the sex-primed chromosome would offer profound insights into the evolutionary dynamics of Schmidtea chromosomes. Clarifying these points would also strengthen the study and enrich the understanding of genome evolution for a wider audience.

Minor comments:

4. The association between heterozygosity and chromosomal inversions, particularly on Schmidtea mediterranea's chromosome 1, is a fascinating aspect of the study. However, the visualization of these inversions in Fig. 1E needs to be clarified (e.g., three inversions are not identifiable). It is recommended that data presentation be improved to make this finding more accessible to readers.

5. The discussion on metazoan ancestral linkage groups (MALG) and the corresponding legends for Fig. 4E to 4G could be clearer. The authors are encouraged to enhance the clarity of these discussions and figure legends, facilitating a better understanding of the evolutionary relationships and genomic structures being presented.

6. Using full species names within the analysis would greatly enhance the manuscript's accessibility and allow readers to fully appreciate the findings, particularly in the data like Fig. 4 and Table 1. This change would help demystify the scientific jargon and make the research more approachable for a broader audience.

7. Integrating a phylogenetic tree into Fig. 4B would provide a much clearer evolutionary context for the genomic data presented. This addition would help readers visualize the phylogenetic relationships and evolutionary history underlying the observed genomic structures.

8. A filtering strategy could reduce the visual clutter shown as gray lines in Fig. 4B, which likely represent gene translocations. This would help minimize visual noise and focus attention on the most relevant genomic relationships and changes.

9. Reordering chromosomes for alignment in Fig. 4D could improve clarity, particularly in interpreting the inversions on chromosome 1. Presenting these inversions as segmental rather than discrete events would provide a clearer understanding of their genomic context and evolutionary significance.

REVIEWER COMMENTS

Reviewer #1 (Remarks to the Author):

First, I would suggest acceptance with minor revisions. This paper is quite good and is not missing much. Only a few comments need to be addressed as listed below.

Thank you!

Comments:

Line 270: 20% of enhancers designated as “distal intergenic” is used as evidence that *S. mediterranea* regulatory elements are close to genes, but there is no evidence presented that 20% is unusual. The paper would benefit from a distal vs proximal comparison to other organisms such that we know if this is highly unusual or not.

You are correct that a comparison to other systems would be desirable. We have initially attempted to compare our findings to regulatory element distributions in Mammals, *Caenorhabditis elegans*, and *Drosophila melanogaster*. However, we had to conclude that our assignment of putative distal enhancers to genes via proximity alone presents a significant roadblock. Therefore, we have modified this section to make it clearer that this is solely a hypothesis at the moment. The section reads:

In addition, the large fraction of intronic enhancers and the fact that only 20% of putative enhancers were classified as “distal intergenic” raises the possibility that regulatory elements in *S. mediterranea* may be tightly associated with genes. Such a tight association between regulatory elements and genes has also been proposed in a tunicate species complex with rapid but chromosome-arm-restricted gene shuffling 78.

In addition, the recent finding that many planarian genes may be regulated by an interplay between AT-rich motifs in the vicinity of the TSS via nucleosome positioning 38 may further contribute to a gene-centric regulatory structure. However, our current association of genes with putative enhancers is based solely on proximity, and systematic functional analyses will be required to demonstrate that planarian genes are indeed more tightly associated with their regulatory elements than in other species.”

Line 276: Given that the whole-animal averaged peak annotations are likely a subset of “constitutively expressed genes or those active in the most abundant cell types. “, is this a sufficient set to support the claim that *S. mediterranea* regulatory elements tend not to be distal? Do constitutively active and highly expressed genes in other species normally have distal enhancers, or are these genes that tend to have local regulatory regions?

Done. The discussion section we added refers to a study investigating the number of enhancers associated with housekeeping genes across mammals and refer to a possible explanation for the difference in Schmidtea. The text in the discussion reads:

“The lack of distal enhancers is striking because our whole-animal averaged peaks annotations are likely enriched for housekeeping genes which are associated with multiple distal enhancers across mammals 1. A possible explanation could be that constitutively expressed genes are regulated by chromatin remodelers, as has recently been suggested for *S. mediterranea* 41.”

Genome Architecture and Synteny: BUSCO is used for gene synteny across species including parasitic flatworms and it is suggested that a clade specific BUSCO data set would be useful as there are 125 missing BUSCOs in all 4 flatworms studied. Given that BUSCO is used for the synteny analysis with parasitic flatworms, it would be nice to know if these missing BUSCOs are also missing in parasitic flatworms.

Done. We have added the BUSCO presence/absence data to the supporting information (TabS5) and refer to it in the text.

Citations: Several bioinformatics tools are missing citations, including trimmomatic, STAR, samtools, SALSA, PICARD, bwa-mem, pbmm2 and DeepVariant. There may be others that were missed. Please check and make sure all tools/contributions are appropriately cited.

Done. We have added these data and references where needed. Additionally, we will provide a GitHub repository containing the code used for the analysis. Thanks for pointing this out.

Methods: No mention is made of which BUSCO lineage was used for the figures or analyses; this should be explicitly stated.

Done. We now provide the information in the Method section; it reads: 'We assessed the BUSCO content of the assemblies using BUSCO (v5.0.0), with the 954 genes in the metazoa_odb10 lineage dataset.'

Reviewer #2 (Remarks to the Author):

Ivankovic et al present the genome assembly and annotation of the flatworm *Schmidtea mediterranea*, and three other sister species in *Schmidtea*, as well as an analysis of gene regulatory regions based on ATAC-seq and ChIP-seq data. They find that promoter- and enhancer-associated histone marks and open chromatin regions are broadly conserved across the genus, and that gene order (synteny) relative to other animals has been largely lost in *Schmidtea*. These findings, in addition to opening new research directions into how flatworm genomes are evolving, will provide a valuable genomics resource to the research community. This is a strong contribution and very interesting manuscript. There are no major flaws; the points below can be considered for further strengthening/clarifying.

Thank you for the positive assessment of our work.

In Fig 1, the authors benchmark the new 'S3BH_all' gene annotations (and others such as the commonly used dd_v6 transcriptome assembly) against various standards such as published transcripts from genbank, BUSCO, etc. It would also be helpful to see a direct comparison to dd_v6, given that so many planarian labs use this particular version. Are the S3BH_all transcripts a superset of the dd_v6 transcripts, or are there some genes that are missing? In cases when they disagree, does it appear there is something wrong with the dd_v6 transcript? An actual list of the disagreements, grouped by type, and some sort of figure panel summarizing the list might be helpful.

We agree that this is an important point. Benchmarking versus dd_v6 has indeed been an important aspect of our annotation effort and dd_v6 is used as a reference in all quality control metrics in Fig. 1. We agree that a gene-by-gene conversion table between the two annotations would be ideal. However, while the majority of annotations in the two data sets have indeed easily identifiable 1:1 mappings, a significant fraction of gene models display 1:many or many:1 mappings. As the examination of the 96 test gene set in Fig. 1 shows, our new annotations outperform v6 by a significant margin, meaning they are better on average. However, they still contain mistakes, as illustrated by the follistatin chimera that we feature in the discussion. Therefore, the resolution of the ambiguous mappings and the correction of the remaining mistakes in the new annotations will require manual curation, which we plan to implement into PlanMine in the medium term. As a stop gap measure, we have now integrated the new annotations into the Rosetta stone tools available at <https://planosphere.stowers.org/> in collaboration with Dr. Eric Ross. This allows the community to cross-compare the annotations on a gene-by-gene basis and to examine ambiguities in either annotation. We hope that this service will help catalyze the acceptance of the new annotations by the community.

One of the fascinating results of this paper is that synteny with the Metazoan Ancestral Linkage Groups (MALGs) has been lost independently in more than one platyhelminth lineage. Based on this the authors suggest that gene positioning might not be important in flatworms. The data seem consistent with this idea. However, I can also imagine a situation where genes cluster on the chromosomes according to their function, and where clustering itself is conserved despite exchangeability of gene position within and between clusters. The authors could examine whether genes cluster spatially in the various species according to their function, for example by looking at Gene Ontology (GO) terms. Alternatively, the authors could limit 'gene positioning may not be essential' to a claim that conservation of synteny is not essential (which the data more strongly indicate).

Done. In fact, we have examined the possibility of functional clustering at various stages of the project, but not obtained convincing signals yet. Therefore, we have implemented your second suggestion and modified the Abstract. It now reads: "Overall, our results suggest that platyhelminth genomes can evolve without syntenic constraints."

Methods Section: I can't find a description of how the whole-genome alignments were done. Did the authors use Progressive Cactus or something like that?

Done. We added technical detail on the alignment process to the Method section. It now reads: "To determine neutral evolutionary distance across Schmidtea, we generated a reference-free whole-genome alignment using Progressive Cactus (v2.6.9 120). Initially, we soft-masked each genome utilizing our custom repeat annotations. Following this, we conducted the whole-genome alignment with the default settings. The output HAL file was then converted to the MAF format using the cactus-hal2maf tool."

AND

"To compare the haplotypes with each other and compare them to the schMedS2 assembly, we aligned them using minimap2 (-asm5, v2.24, 97) and parsed the alignments using syri (v1.6.3, 98)."

Data Access Section: Please remember to make the repetitive element annotations publicly available in gff format or equivalent.

Done. We have uploaded the annotations to a Zenodo archive (DOI: [10.5281/zenodo.11004547](https://doi.org/10.5281/zenodo.11004547)), where it will be made available upon publication.

Figure 1A: The lack of long-range contacts in the Hi-C data is curious, so readers might benefit from some more commentary about this in the discussion section. Guo et al (2022) saw something similar in their Schmidtea Hi-C data, so evidently it is reproducible, but I don't see much commentary except on lines 542-543. How common is this phenomenon in other species? Please add a reference or two.

Done. We have added a brief discussion and a few references. The text now reads:

“One of the interesting questions raised by our findings is how flatworms can achieve gene expression specificity, apparently without the topological constraints that are important in other systems 15–21. An indication that planarians may achieve gene expression specificity by unusual means is that topologically associated domains (TADs) are not apparent in our Hi-C data (Fig. 1A, Fig. 3B), and although A/B compartments can be called in *S. mediterranea*, they do not show sharp boundaries or the typical partition into gene-rich and gene-poor compartments³⁸. Thus, planarians appear more similar to cnidarians⁷⁴, which lack TADs or *C. elegans*, where TADs on the autosomes are less pronounced⁷⁶ and classic TADs are restricted to the X chromosome⁷⁷.”

Figure 1B: Mercury spectra-cn plots should be added to the supplement. Some things like spurious duplication levels are not (as I understand it) captured in the quality value or completeness metrics alone and it can be diagnostic to look directly at the spectra-cn. Not necessarily for all four read data sets, but perhaps a representative one or two.

Done. We have added the cn-spectra plots for one short-read dataset to the supporting information under Section 1.1. and refer to it from the text.

Figure 1H: On the x-axis the label “dd_v1.PCF” is repeated and “dd_v6.PCF” is missing, so one of the bars must be mis-labeled.

Done. The error has been corrected. Thanks for pointing this out.

Figure 2E: A Venn Diagram seems more appropriate than a bar chart, especially for the rightmost bar, which requires an awkward sort of legend to explain (which is too small to be read easily).

Done. We have converted the bar charts to pie charts and increased the label size.

Lines 270: “indicates” could be changed to “suggests”. Given the data, it’s a reasonable hypothesis that regulatory elements are close to genes in *Schmidtea*, but more data are probably needed to use “indicates”.

Done. The text now reads: “In addition, the large fraction of intronic enhancers, the fact that only 20% of putative enhancers were designated as “distal intergenic” and that all identified H3K27ac peaks had a median distance of 1693bp to the TSS suggests that regulatory elements in *S. mediterranea* might be particularly closely associated with genes.”

Figure 2G: Do these proportions look different if the enhancers/promoters are stratified according to whether or not their associated gene is variably expressed across tissues in the planarian single-cell sequencing atlas? Housekeeping genes usually comprise a sizeable chunk of the transcriptome, and I’d suppose they’d also show strong ChIP-seq peaks, so any tissue-specific signal might be obscured. If you need more room to extend 2G to accommodate this, I’d remove 2I, which seems redundant.

Thank you for the suggestion. Full integration of the gene annotation with the available scRNA-seq and other publicly available data is still a work in progress and beyond the scope of this work. It will be included in a future release of PlanMine, the genome resources platform our lab maintains.

Figure 2H-I and Figure 3H: The x-axis should be labeled in kilobases relative to the TSS in these plots.

We chose to retain the absolute coordinates of these regions to facilitate inspection in the genome browser. The distance to the TSS is given in the text for Figure 2H-I.

Figure 3H: The TSS of the *S. mediterranea* ATAC-seq plot should be aligned with the TSS of the three sister species in the plots below.

Done.

Line 288: 'data not shown': Perhaps there is a reference that can be added or some other way of saying this.

Done. We have adjusted the text by removing references to unpublished data.

Figure 4B: The chromosome order and orientations in the riparian plot should be chosen to minimize the crossing of 'braids' between all species. This will make it easier to understand the evolutionary history. I believe it can be done by increasing the 'syntenyWeight' parameter in the 'plot_riparian()' function of GENESPACE. For instance, crossing is decreased by making the *S. lugubris* chromosome order 1-3-2-4 instead of 1-2-3-4.

Thank you for the suggestion. We have attempted to further reorder the chromosome using this function but were not successful. However, together with the other changes to Figure 4 that we describe below, we do not think it is necessary to make any further rearrangements.

Figure 4H-J: Based on Figure 4D-G, it seems clear that there was independent loss of synteny relative to the Metazoan Ancestral Linkage Groups (MALGs) in both Schmidtea and Macrostomorpha, so in Figure 4H-J the chosen outgroup (*M. hystrix*) is derived, which is not ideal in terms of clarity and ease of interpretation. Could either choose an outgroup for panels H-J that is ancestral for synteny with other animals (perhaps some mollusk?) or move H-J to the supplement (it seems redundant with panels E-G anyway, so maybe the latter is the way to go).

This point seems to be a misunderstanding. The aim of Figure 4H-J is to show the loss of synteny between the flatworm groups. In response to your and reviewer 3's suggestion, we have now separated Figure 4H-J into a new figure to allow better readability and the addition of silhouettes to enhance clarity. We have adjusted the legend of Figure 5 as well; it now reads: "Detection of metazoan ancestral linkage groups (MALG) and pairwise synteny in three flatworms species. [...] No chromosome combination was

enriched except for 57 orthologs between *S. mansoni* and *S. mediterranea* located on Chromosome 6 and Chromosome 4, respectively), indicating that synteny between these three flatworm groups is not conserved.”

Typographical points:

Lines 211-213: The sentence beginning with “Collectively...” has confusing syntax, and maybe also a typo (“93 out of 91”?). Perhaps it should read: “Collectively, the S3 annotations had the highest fraction of error-free gene models, contained models of all the test genes, and had intact ORF representations for 93 out of 96 test genes.”

Done. We have reworded the sentence it now reads:

“Overall, the S3 annotations performed best, containing models of all test genes and the highest proportion of error-free gene models with intact ORF representations of 93/96 test genes. “

Line 271: Gene name “sp5” should be italicized.

Done.

Line 286: "Namely, all known members of the genus, " Replace with a colon (sentence fragment).

Done. We have adjusted the sentence it now reads:

“...we sequenced the genomes of *S. mediterranea*’s three closest relatives, *S. polychroa*, *S. nova*, and *S. lugubris*”

Line 316: “robust” should be “robustness”.

Done.

Line 450: “significantly improves over” should be “is a significant improvement over” (improve should not be used intransitively here).

Thank you for pointing this out. We have reworded the section to streamline the text, and the term significantly is now not used anymore.

Line 550: “artefect” should be “artifact”.

Done.

Reviewer #3 (Remarks to the Author):

In this manuscript, Ivankovic and Brand et al. present a haplotype-phased genome assembly for a sexually reproducing strain of *Schmidtea mediterranea*, alongside chromosome-scale genome sequences for three related *Schmidtea* species. Over recent decades, *S. mediterranea* has become a famous model system for stem cell

biology. However, it has been challenging to sequence its genome due to its unique biology, such as high AT and repeat content. As a result, the first high-quality genome for a sexually reproducing *S. mediterranea* strain was published in 2018, utilizing long-read sequencing. Subsequently, in 2022, another sexual strain was sequenced to chromosome-scale by the other group using advanced Hi-C scaffolding. This study showed that *S. mediterranea* possesses four chromosomes, with chromosome 1 suggested as a potential sex-primed autosome due to its reduced recombination rate.

This study builds upon previous work by resequencing the *S. mediterranea* genome and sequencing three additional species within the *Schmidtea* genus. In addition, the authors employ ATAC-seq and ChIP-seq to explore the evolution of epigenetic regulation in planarians. The authors discover that the reduced recombination rate in chromosome 1 of *S. mediterranea* is attributable to several large inversions that distinguish between haplotypes 1 and 2. They further conducted comparative analyses with other flatworms to explore the conservation of genome regulation and structure.

The manuscript is commended for its clarity and the breadth of its findings, appealing to a broad audience. I particularly appreciate the effort to contextualize the study within the wider classification of planarians as part of the Platyhelminthes phylum. Despite the widespread recognition of *S. mediterranea* as a model, it is essential to remember that it belongs to a rapidly evolving, derived lineage within the flatworms, marked by distinct biological characteristics and lineage-specific traits.

Thank you for your appreciation of our effort.

Nonetheless, the current manuscript could be improved by enhancing the comparative genomics and synteny analyses.

Major comments:

1. The manuscript is a pioneering work in comparative genomics within the *Schmidtea* genus. Yet, it appears to overlook the inclusion of a closely related outgroup and the reconstruction of ancestral features, essential elements of comparative genomics. Incorporating a closely related outgroup would significantly enhance the comparative analysis, offering more profound insights into the genomics of *Schmidtea*. In instances where chromosome-scale genomic data may not clarify phylogenetic relationships, considering draft genomes and transcriptome data previously published by the authors could provide valuable additional context.

We fully agree with the reviewer about the utility of outgroup analyses regarding the assessment of the ancestral state or the direction of evolutionary change. Unfortunately, there are no useful data sets available for providing an outgroup perspective on *Schmidtea* chromosome evolution as the one point of the manuscript that would clearly benefit from the inclusion of an outgroup. The two genome assemblies for the planarian *Dugesia japonica* as, to our knowledge, only other publicly available planarian genome assemblies, are too fragmented for an assessment of chromosome structure (N50 values of 27,741 bp and 248.44 kb). Therefore, our

analysis includes all relevant and available data. However, we fully agree with the reviewer about the many interesting questions that could be addressed by having more end-to-end planarian genome assemblies and we are looking forward to incorporating them into our analysis, once they become available.

2. The manuscript would benefit from a clearer explanation regarding which Schmidtea genomes reflect the ancestral state in the synteny analysis, particularly concerning the evolution of the sex-primed chromosome in *S. mediterranea*. Clarifying whether this chromosome resulted from the fusion of chromosomes from different Schmidtea lineages would significantly enhance the narrative and findings. This leads to the following comment.

Including a chromosome-scale genome assembly of a closely related outgroup would indeed be interesting. Given the current lack of such an assembly (see above), we would like to refrain from a formal ancestral state reconstruction without such resources. We have added a sentence in the discussion to that effect:
" Similarly, we find that the *S. mediterranea* Chromosome 1 synteny block was split into two parts in the other three species, suggesting its origin in a fusion with mixing event."

3. In the section starting around Line 384, the writing leaves some ambiguity regarding the occurrence of 'split' or 'fusion' events in the chromosomes being discussed. As previously mentioned, this uncertainty likely arises from a notable absence of detailed ancestral state reconstruction. For example, a cursory review of Fig. 4B reveals at least seven linkage groups in the ancestral state across all Schmidtea genomes examined. In fact, the detection of gene intermingling between schMedS3h2 chromosome 1 and schPol2 chromosomes 3 and 4 suggests a fusion-with-mixing event may have occurred from schPol2 to schMedS3h2, highlighting a schMed-specific fusion event. This observation supports the notion that fusion-with-mixing events, often considered irreversible and directional, play a crucial role in chromosomal evolution (e.g., lineage-specific fusions on chromosomes 1, 2, and 3 unique to schMed). Elucidating the impact of these events on the development of the sex-primed chromosome would offer profound insights into the evolutionary dynamics of Schmidtea chromosomes. Clarifying these points would also strengthen the study and enrich the understanding of genome evolution for a wider audience.

As stated above, we believe that the analysis of Schmidtea chromosome evolution will require additional genomes. But we have added a sentence in the discussion mentioning the interesting hypothesis raised by the reviewer (thank you for this). It reads: ". Similarly, we find that the *S. mediterranea* Chromosome 1 synteny block was split into two parts in the other three species, suggesting its origin in a fusion with mixing event."

Minor comments:

4. The association between heterozygosity and chromosomal inversions, particularly on Schmidtea mediterranea's chromosome 1, is a fascinating aspect of the study.

However, the visualization of these inversions in Fig. 1E needs to be clarified (e.g., three inversions are not identifiable). It is recommended that data presentation be improved to make this finding more accessible to readers.

Done. We have adapted Fig. 1E to clearly indicate the two smaller inversions contained in the large inversion on Chromosome 1.

5. The discussion on metazoan ancestral linkage groups (MALG) and the corresponding legends for Fig. 4E to 4G could be clearer. The authors are encouraged to enhance the clarity of these discussions and figure legends, facilitating a better understanding of the evolutionary relationships and genomic structures being presented.

Done. We have split Fig. 4 to allow for better readability by adding silhouettes and better labels. We have also revised the figure legend of the new Fig. 5. It now reads: "Detection of metazoan ancestral linkage groups (MALG) and pairwise synteny in three flatworms species. [...] No chromosome combination was enriched except for 57 orthologs between *S. mansoni* and *S. mediterranea* located on Chromosome 6 and Chromosome 4, respectively), indicating that synteny between these three flatworm groups is not conserved."

6. Using full species names within the analysis would greatly enhance the manuscript's accessibility and allow readers to fully appreciate the findings, particularly in the data like Fig. 4 and Table 1. This change would help demystify the scientific jargon and make the research more approachable for a broader audience.

Done and thank you for this suggestion. Fig. 4 has been split into two and both the new Fig. 4 and Fig.5 now use species names instead of the assembly names. We have also added an additional row to Table 1 giving the species name.

7. Integrating a phylogenetic tree into Fig. 4B would provide a much clearer evolutionary context for the genomic data presented. This addition would help readers visualize the phylogenetic relationships and evolutionary history underlying the observed genomic structures.

Done. We have added a cladogram to Fig. 4B and Fig. 4E to allow for easy placement of the shown rearrangements in an evolutionary context.

8. A filtering strategy could reduce the visual clutter shown as gray lines in Fig. 4B, which likely represent gene translocations. This would help minimize visual noise and focus attention on the most relevant genomic relationships and changes.

These larger synteny blocks that appear to be translocated could either reflect true translocations, misidentification of orthology, or assembly artifacts. We prefer to include these to be fully transparent about the weight of evidence. However, we have increased the size of Fig. 4B which makes it much easier to read the panel.

9. Reordering chromosomes for alignment in Fig. 4D could improve clarity, particularly in interpreting the inversions on chromosome 1. Presenting these inversions as segmental rather than discrete events would provide a clearer understanding of their genomic context and evolutionary significance.

We have enhanced Fig. 1 to better reflect the segmental inversion on Chromosome 1. We have also experimented with various representations of the inversion in Fig. 4 but found them to be visually confusing. Finally, with the other changes to Figure 4, we do not think it is necessary to make any further rearrangements.

REVIEWERS' COMMENTS

Reviewer #2 (Remarks to the Author):

The authors have done an excellent job of addressing reviewer comments, and in my view the manuscript is now suitable for publication. This will be a very nice contribution to the field.

Reviewer #3 (Remarks to the Author):

The authors have thoroughly addressed all my comments. While including an outgroup would provide more insight into the chromosome evolution of planarians, it is understandable that additional chromosome-level genomes are necessary. The authors have also clarified their data presentation, which should be commended. I have no further concerns and recommend publishing this important study.